# Molecular mimicry in multisystem inflammatory syndrome in children

Aaron Bodansky[1,35], Robert C. Mettelman[2,35], Joseph J. Sabatino Jr[3,4], Sara E. Vazquez[5], Janet Chou[6,7], Tanya Novak[8,9], Kristin L. Moffitt[7,10], Haleigh S. Miller[5,11], Andrew F. Kung[5,11], Elze Rackaityte[5], Colin R. Zamecnik[3,4], Jayant V. Rajan[5], Hannah Kortbawi[5,12], Caleigh Mandel-Brehm[5], Anthea Mitchell[13], Chung-Yu Wang[13], Aditi Saxena[13], Kelsey Zorn[5], David J. L. Yu[14], Mikhail V. Pogorelyy[2], Walid Awad[2], Allison M. Kirk[2], James Asaki[15], John V. Pluvinage[4], Michael R. Wilson[3,4], Laura D. Zambrano[16], Angela P. Campbell[16], Overcoming COVID-19 Network Investigators*, Paul G. Thomas[2,36], Adrienne G. Randolph[7,8,9,36], Mark S. Anderson[14,17,36 ✉] & Joseph L. DeRisi[5,13,36 ✉]

Multisystem inflammatory syndrome in children (MIS-C) is a severe, post-infectious sequela of SARS-CoV-2 infection[1,2], yet the pathophysiological mechanism connecting the infection to the broad inflammatory syndrome remains unknown. Here we leveraged a large set of samples from patients with MIS-C to identify a distinct set of host proteins targeted by patient autoantibodies including a particular autoreactive epitope within SNX8, a protein involved in regulating an antiviral pathway associated with MIS-C pathogenesis. In parallel, we also probed antibody responses from patients with MIS-C to the complete SARS-CoV-2 proteome and found enriched reactivity against a distinct domain of the SARS-CoV-2 nucleocapsid protein. The immunogenic regions of the viral nucleocapsid and host SNX8 proteins bear remarkable sequence similarity. Consequently, we found that many children with anti-SNX8 autoantibodies also have cross-reactive T cells engaging both the SNX8 and the SARS-CoV-2 nucleocapsid protein epitopes. Together, these findings suggest that patients with MIS-C develop a characteristic immune response to the SARS-CoV-2 nucleocapsid protein that is associated with cross-reactivity to the self-protein SNX8, demonstrating a mechanistic link between the infection and the inflammatory syndrome, with implications for better understanding a range of post-infectious autoinflammatory diseases.

Children with severe acute respiratory syndrome coronavirus 2 (SARS-CoV-2) infections typically have mild disease[3,4], but can develop a rare life-threatening post-infectious complication known as MIS-C[1,2]. MIS-C presents with a distinctive inflammatory signature indicative of altered innate immune responses[5,6], including dysregulation of the mitochondrial antiviral signalling (MAVS) protein pathway[7]. Aberrant adaptive immunity is also involved, with multiple MIS-C-associated autoantibodies reported[8–12]. Furthermore, T cell signatures have also been associated with development of MIS-C[13–16], which are accompanied by autoimmune-associated B cell expansions[8]. Some autoimmune diseases have been shown to involve tandem cross-reactive B cell and T cell responses. In multiple sclerosis, for example, cross-reactive

B cells and T cells have been shown to respond to Epstein–Barr virus protein (EBNA1) and antigens in the human nervous system[17–19]. Decades of research into paraneoplastic autoimmune encephalitis has also demonstrated that autoreactive B cells and T cells can cause disease through coordinated targeting of a shared intracellular antigen and, in certain cases, a shared epitope[20–26]. Despite intense interest, a pathophysiological link between SARS-CoV-2 and MIS-C remains enigmatic, and identification of disease-specific autoantigens remains incompletely explored. Here children previously infected with SARS-CoV-2 with (n = 199) and without (n = 45) MIS-C were enrolled and comprehensively evaluated for differential autoreactivity to the entire human and SARS-CoV-2 proteome. Patients with MIS-C were found to have

[1]Department of Pediatrics, Division of Critical Care, University of California San Francisco, San Francisco, CA, USA. [2]Department of Host–Microbe Interactions, St. Jude Children's Research Hospital, Memphis, TN, USA. [3]Weill Institute for Neurosciences, University of California San Francisco, San Francisco, CA, USA. [4]Department of Neurology, University of California San Francisco, San Francisco, CA, USA. [5]Department of Biochemistry and Biophysics, University of California San Francisco, San Francisco, CA, USA. [6]Division of Immunology, Department of Pediatrics, Boston, MA, USA. [7]Department of Pediatrics, Harvard Medical School, Boston, MA, USA. [8]Department of Anesthesiology, Critical Care and Pain Medicine, Boston Children's Hospital, Boston, MA, USA. [9]Department of Anesthesia, Harvard Medical School, Boston, MA, USA. [10]Department of Pediatric, Division of Infectious Diseases, Boston Children's Hospital, Boston, MA, USA. [11]Biological and Medical Informatics Program, University of California San Francisco, San Francisco, CA, USA. [12]Medical Scientist Training Program, University of California San Francisco, San Francisco, CA, USA. [13]Chan Zuckerberg Biohub SF, San Francisco, CA, USA. [14]Diabetes Center, School of Medicine, University of California San Francisco, San Francisco, CA, USA. [15]Biomedical Sciences Program, University of California San Francisco, San Francisco, CA, USA. [16]COVID-19 Response Team and Coronavirus and Other Respiratory Viruses Division, Centers for Disease Control and Prevention, Atlanta, GA, USA. [17]Department of Medicine, Division of Endocrinology and Metabolism, University of California San Francisco, San Francisco, CA, USA. [35]These authors contributed equally: Aaron Bodansky, Robert C. Mettelman. [36]These authors jointly supervised this work: Paul G. Thomas, Adrienne G. Randolph, Mark S. Anderson, Joseph L. DeRisi. *A list of authors and their affiliations appears at the end of the paper. ✉e-mail: mark.anderson@ucsf.edu; joe@derisilab.ucsf.edu

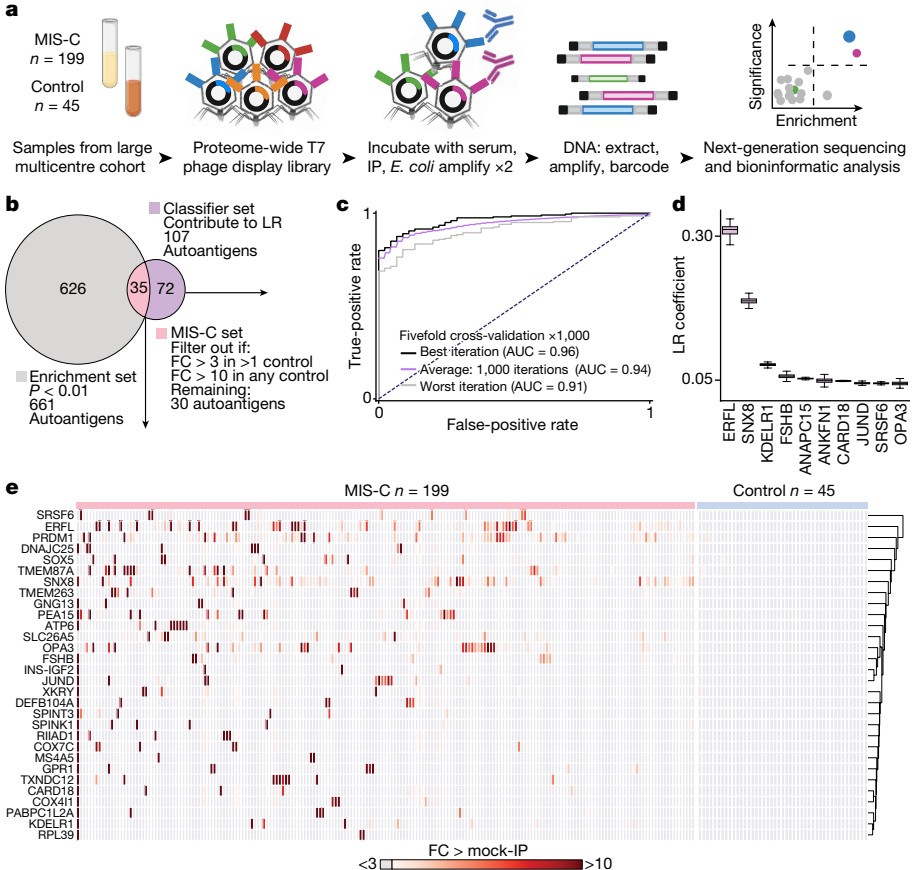

**Fig. 1 | Autoantigens distinguish MIS-C from at-risk controls. a**, Design of the PhIP-seq experiment comparing patients with MIS-C (*n* = 199) and at-risk controls (*n* = 45; children with SARS-CoV-2 infection at least 5 weeks before sample collection without symptoms of MIS-C). Schematics in panel **a** were created using BioRender (https://www.biorender.com). **b**, Venn diagram highlighting the number of autoantigens identified with statistically significant PhIP-seq enrichment ('enrichment set': grey circle; *P* < 0.01 on one-sided Kolmogorov–Smirnov test with false discovery rate correction) and autoantigens identified, which contribute to a logistic regression classifier of MIS-C relative to at-risk controls ('classifier set': purple circle). There are 35 autoantigens present in both the classifier set and the enrichment set (pink; union of the Venn diagram) of which 30 are exclusive to MIS-C and referred to as the 'MIS-C set'

(no two controls have low reactivity as defined by the fold-change (FC) signal over the mean of protein A/G beads only (FC > mock-IP) of 3 or greater, and no single control has high reactivity defined as FC > mock-IP greater than 10). LR, logistic regression. **c**, Receiver operating characteristic curve for the logistic regression classifier showing upper and lower bounds of performance through 1,000 iterations. **d**, Bar plots with error bars showing logistic regression coefficients for the top 10 autoantigens across 1,000 iterations. The whiskers extend to 1.5 times the interquartile range (IQR) from the quartiles. The boxes represent the IQR, and the centre lines represent the median. **e**, Hierarchically clustered (Pearson) heatmap showing the PhIP-seq enrichment (FC > mock-IP) for the 30 autoantigens in the MIS-C set in each patient with MIS-C and each at-risk plasma control.

both cross-reactive antibodies and T cells targeting an epitope motif shared by the viral nucleocapsid protein and human SNX8, a protein involved in MAVS antiviral function[27]. These findings suggest that many cases of MIS-C may be triggered by molecular mimicry and could provide a framework for identifying potential cross-reactive epitopes in other autoimmune and inflammatory diseases with predicted viral triggers such as Kawasaki disease[28], type 1 diabetes mellitus (T1DM)[29] and multiple sclerosis.

## Patients with MIS-C have a distinct set of autoreactivities

To explore the hypothesis that MIS-C is driven by an autoreactive process, we evaluated the proteome-wide autoantibody profiles of children with MIS-C (*n* = 199) and children convalescing following asymptomatic or mild SARS-CoV-2 infection without MIS-C (*n* = 45, hereafter referred to as 'at-risk controls') using our custom phage immunoprecipitation and sequencing (PhIP-seq)[30] library, which has previously been used to define novel autoimmune syndromes and markers of disease for various conditions[12,24,25,31–33]. Given the inherently heterogeneous

nature of antibody repertoires among individuals[34], the identification of disease-associated autoreactive antigens requires the use of large numbers of cases and controls[12]. To minimize spurious hits, this study includes substantially more patients with MIS-C and controls than similar, previously published studies[8–10,12] (Fig. 1a). Clinical characteristics of this cohort are described in Extended Data Table 1.

For a given set of samples, PhIP-seq can yield dozens to thousands of differential enrichments of phage-displayed peptides. Here logistic regression machine learning was used as an initial unbiased measure of how accurately a set of differentially enriched peptides could classify people with MIS-C and controls—an approach that has been used to classify people with autoimmune polyglandular syndrome type 1 using PhIP-seq data[12]. In all, 107 proteins had logistic regression coefficients greater than zero ('classifier set'; Fig. 1b). As this is an unbalanced dataset with a random accuracy less than 50%, we also generated a receiver operating characteristic (ROC) curve. ROC analysis iterated 1,000 times and yielded an average area under the curve (AUC) of 0.94 (Fig. 1c). Examination of the logistic regression coefficients associated with MIS-C revealed the largest contributions from peptides derived from the ETS repressor factor-like (ERFL), sorting nexin 8 (SNX8) and

KDEL endoplasmic reticulum protein retention receptor 1 (KDELR1) coding sequences (Fig. 1d).

In parallel, a Kolmogorov–Smirnov test was used to define a set of 661 autoreactivities statistically enriched after false discovery rate adjustment for multiple comparisons ($q < 0.01$; 'enrichment set'). To avoid false positives, the intersection of the classifier set and enrichment set were considered further. Of these 35 hits, peptides derived from 30 different proteins satisfied an additional set of conservative criteria, requiring that none was enriched (fold change over mock-immunoprecipitation (IP) of more than 3) in more than a single control, or was enriched more than 10-fold in any control ('MIS-C set'; Fig. 1e).

## Previously reported MIS-C autoantibodies

To date, at least 34 autoantigen candidates have been reported to associate with MIS-C[8–10,12]. However, we found that only UBE3A (a ubiquitously expressed ubiquitin protein ligase) was differentially enriched in our MIS-C dataset, whereas the remaining 33 were present in a similar proportion of cases with MIS-C and at-risk controls (Extended Data Fig. 1a). Autoreactivity to UBE3A was independently identified in this study as part of both the classifier and the enrichment sets, but was not included in the final MIS-C set due to the low positive signal present in two controls.

In addition, autoantibodies to the receptor antagonist IL-1RA have been previously reported in 13 of 21 (62%) patients with MIS-C[11]. In this cohort, anti-IL-1RA antibodies were detected by PhIP-seq ($z$ score > 6 over at-risk control) in six patient samples. To further examine immune reactivity to full-length IL-1RA, sera from 196 of the 199 patients in this study were used to immunoprecipitate [35S]-methionine-radiolabelled IL-1RA (radioligand-binding assay (RLBA)). Positive immunoprecipitation of IL-1RA (defined as more than 3 s.d. above mean of controls) was found in 39 of 196 (19.9%) patients with MIS-C. However, many patients with MIS-C were treated with intravenous immunoglobulin (IVIG), a blood product shown to contain autoantibodies[35]. After removing samples from patients treated with IVIG (61 remaining), the difference between samples from patients with MIS-C (5 of 61, 8.2%) and at-risk controls (1 of 45, 2.2%) was not significant ($P = 0.299$; Extended Data Fig. 1b).

## MIS-C autoantigens lack tissue-specific associations with clinical phenotypes

Consistent with previous MIS-C reports[1,5], this cohort was clinically heterogeneous (Extended Data Table 2). To determine whether specific phenotypes, including myocarditis and the requirement of vasopressors, might be associated with specific autoantigens present in the MIS-C set, tissue expression levels were assigned to each autoantigen[36] (Human Protein Atlas; https://proteinatlas.org), including the amount of expression in cardiomyocytes and the cardiac endothelium. The PhIP-seq signal for patients with MIS-C with a particular phenotype was compared with those patients with MIS-C without the phenotype. Autoantigens with tissue specificity were not enriched in those patients with MIS-C with phenotypes involving said tissue. Similarly, autoantigens associated with myocarditis or vasopressor requirements did not correlate with increased cardiac expression (Extended Data Fig. 1c).

## Orthogonal validation of PhIP-seq autoantigens

Peptides derived from ERFL, SNX8 and KDELR1 carried the largest logistic regression coefficients in the MIS-C classifier. The PhIP-seq results were orthogonally confirmed by RLBAs using full-length ERFL, SNX8 and KDELR1 proteins. Relative to at-risk controls, samples from patients with MIS-C significantly enriched each of the three target proteins ($P < 1 \times 10^{-10}$ for ERFL, SNX8 and KDELR1), consistent with the PhIP-seq assay (Extended Data Fig. 2a). Using only the RLBA data for

these three proteins, MIS-C could be confidently classified (ROC with fivefold cross-validation; 1,000 iterations) from at-risk control sera with an AUC of 0.93, suggesting the potential for molecular diagnostic purposes (Extended Data Fig. 2b).

As noted, IVIG was administered to 138 of the 199 patients with MIS-C before sample collection and was absent from all 45 at-risk controls. The autoreactivity to the ERFL, SNX8 and KDELR1 proteins from the 61 patients with MIS-C who had not been treated with IVIG before sample collection were compared with the at-risk controls. In contrast to IL-1RA, the differential enrichment of these three proteins remained significant ($P = 6.69 \times 10^{-10}$, $P = 6.26 \times 10^{-5}$ and $P = 0.0001$, respectively), suggesting that autoreactivity to ERFL, SNX8 and KDELR1 proteins was not confounded by IVIG treatment (Extended Data Fig. 2c).

## Independent MIS-C cohort validation

To further test the validity of these findings, an independent validation cohort consisting of samples from 24 different patients with MIS-C and 29 children with severe acute COVID-19 was evaluated (acquired via ongoing enrolment of the Overcoming COVID-19 study; Extended Data Table 3). Using RLBAs with full-length ERFL, SNX8 and KDELR1 proteins, we found that all three target proteins were significantly enriched compared with both the at-risk controls ($P = 0.00022$, $P = 3.68 \times 10^{-5}$ and $P = 2.36 \times 10^{-5}$, respectively) and the patients with severe acute COVID-19 ($P = 0.0066$, $P = 0.00735$ and $P = 0.00114$, respectively; Extended Data Fig. 2d). A logistic regression model, trained on the original cohort, classified MIS-C from at-risk controls with an AUC of 0.84, and from severe acute paediatric COVID-19 with an AUC of 0.78 (Extended Data Fig. 2e). This suggests that autoreactivity to ERFL, SNX8 and KDELR1 is a significant feature of MIS-C that is separable from SARS-CoV-2 exposure and severe acute paediatric COVID-19.

## MIS-C autoantibodies target a single epitope within the SNX8 protein

SNX8 is a protein that is 456 amino acids and belongs to a family of sorting nexins involved in endocytosis, endosomal sorting and signalling[37]. Publicly available expression data[36] (Human Protein Atlas) show that *SNX8* is widely expressed across various tissues including the brain, heart, gastrointestinal tract, kidneys and skin, with the highest expression in undifferentiated cells and immune cells. Previous work has associated SNX8 with host defence against RNA viruses[27]. ERFL is a poorly characterized 354-amino acid protein. A survey of single-cell RNA sequencing (scRNA-seq) data[36] (Human Protein Atlas) suggests enrichment in plasma cells, B cells and T cells in some tissues. Using a Spearman correlation in principal component analysis (PCA) space based on tissue RNA-seq data[36] (Human Protein Atlas), *SNX8* has the second closest expression pattern to *ERFL* compared with all other coding genes, with a correlation coefficient of 0.81. KDELR1 is a 212-amino acid endoplasmic reticulum–Golgi transport protein essential to lymphocyte development with low tissue expression specificity. All three proteins are predicted to be intracellular, suggesting that putative autoantibodies targeting these proteins are unlikely to be sufficient for disease pathology on their own. However, autoantibodies targeting intracellular antigens are often accompanied by autoreactive T cells specific for the protein from which that antigen was derived, and which targets cell types expressing the protein[22,25,26,38]. We selected SNX8 for further investigation, given its enrichment in immune cells and its putative role in regulating the MAVS pathway in response to RNA virus infection, a pathway implicated in MIS-C pathology[7].

Full-length SNX8 is represented in this PhIP-seq library by 19 overlapping 49-mer peptides. For all but one patient sample, the peptide fragment spanning amino acid positions 25–73 was the most enriched in the PhIP-seq assay (Fig. 2a), suggesting a common autoreactive site. A sequential alanine scan was performed to determine the minimal

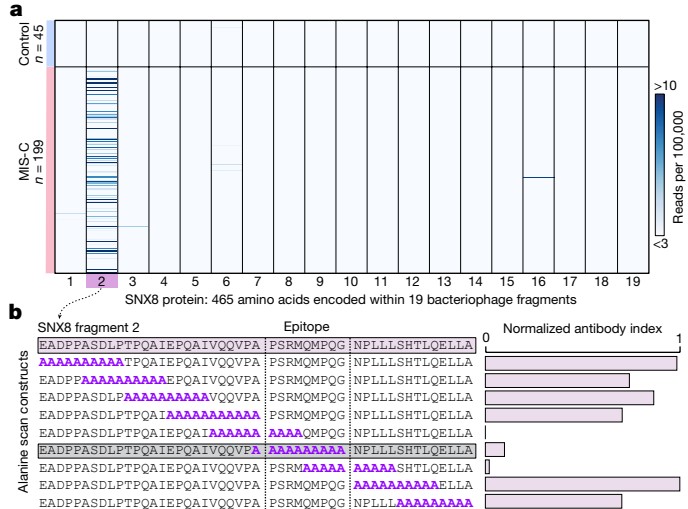

**Fig. 2 | Autoantibodies in patients with MIS-C target a single epitope within SNX8. a**, PhIP-seq signal (reads per 100,000) for each patient with MIS-C (*n* = 199) and each at-risk control (*n* = 45) across each of the 19 bacteriophage-encoded peptide fragments, which together tile the full-length SNX8 protein. **b**, SLBA enrichments (normalized antibody indices) for each sequential alanine mutagenesis construct. Constructs were designed with 10 amino acid alanine windows (highlighted in purple) shifted by 5 amino acids until the entire immunodominant SNX8 region (SNX8 fragment 2) was scanned. Values are averages of six separate patients with MIS-C. The identified autoantibody epitope is bounded by vertical grey dotted lines.

immunoreactive peptide sequence (Fig. 2b; Methods). Using samples from six individuals with MIS-C, we determined that the critical region for immunoreactivity was a nonamer spanning positions 51–59 (PSRMQMPQG). Using the wild-type 49-amino acid peptide and the version with the critical region mutated to alanine, 182 of the 199 patients with MIS-C (insufficient sample for the remaining 17) and all 45 controls were assessed for immunoreactivity using a split-luciferase-binding assay (SLBA). We found that samples from 31 of 182 (17.0%) patients with MIS-C immunoprecipitated the wild-type fragment. Of these, 29 (93.5%) failed to immunoprecipitate the mutated peptide, suggesting a common shared autoreactive epitope among nearly all of the patients with MIS-C with anti-SNX8 antibodies (Extended Data Fig. 2f).

## Patients with MIS-C have an altered antibody response to the SARS-CoV-2 nucleocapsid protein

To evaluate whether differences exist in the humoral immune response to SARS-CoV-2 infection in patients with MIS-C relative to at-risk controls, we repeated PhIP-seq with 181 of the original 199 patients with MIS-C and all 45 of the at-risk controls using a previously validated library specific for SARS-CoV-2 (ref. 39). To discover whether certain fragments were differentially enriched in either patients with MIS-C or at-risk controls, the enrichment of each phage encoded SARS-CoV-2 peptide (38 amino acids each) across all patients with MIS-C and at-risk controls was normalized to 48 healthy controls pre-COVID-19. Three nearly adjacent peptides derived from the SARS-CoV-2 nucleocapsid protein (fragments 5, 8 and 9) were significantly enriched (Kolmogorov–Smirnov test *P* < 0.0001 for each). The first peptide (fragment 5), spanning amino acids 77–114, was significantly enriched in the at-risk controls (representing the typical serological response in children), whereas the next two fragments (fragments 8 and 9), spanning amino acids 134–190, were significantly enriched in patients with MIS-C (Fig. 3a,b). The most differentially reactive region of the SARS-CoV-2 nucleocapsid protein in patients with MIS-C (fragment 8) was termed the MIS-C-associated domain of SARS-CoV-2 (MADS). The PhIP-seq

results were orthogonally confirmed using an SLBA measuring the amount of MADS peptide immunoprecipitated with samples from 16 individuals, including 11 patients with MIS-C and 5 at-risk controls (Fig. 3c). To precisely map the minimal immunoreactive region of MADS in MIS-C samples, peptides featuring a sliding window of ten alanine residues were used as the immunoprecipitation substrate for SLBAs, run in parallel with the SNX8 alanine scanning peptides using sera from three patients with MIS-C (Fig. 3d). The critical regions identified here in both SNX8 and MADS were highly similar, represented by the (ML)Q(ML)PQG motif (Fig. 3e).

## Patients with MIS-C have significantly increased SNX8 autoreactive T cells

In other autoimmune diseases, autoantibodies often arise to intracellular targets, yet the final effectors of cellular destruction are autoreactive T cells[22,26,40]. Given evidence that certain subsets of MIS-C are associated with HLA[16], and that SNX8 is an intracellular protein, we hypothesized that patients with MIS-C with anti-SNX8 antibodies may, in addition to possessing SNX8 autoreactive B cells, also possess autoreactive T cells targeting SNX8-expressing cells. To test this hypothesis, T cells from nine patients with MIS-C (eight from SNX8 autoantibody-positive patients and one who was SNX8 autoantibody negative) and ten at-risk controls (chosen randomly) were exposed to a pool of 15-mer peptides with 11-amino acid overlaps tiling the full-length human SNX8 protein. T cell activation was measured by an activation-induced marker assay, which quantifies upregulation of three cell activation markers: OX40, CD69 and CD137 (ref. 41). The percent of T cells activated in response to SNX8 protein was significantly higher in patients with MIS-C than in controls (*P* = 0.00126). Using a positive cut-off of 3 s.d. above the mean of the controls, 7 of the 9 (78%) patients with MIS-C were positive for SNX8-expressing autoreactive T cells, whereas 0 of 10 (0%) controls met these criteria (Fig. 4a). With respect to CD4+ and CD8+ subgroups, there was an increased signal in patients with MIS-C compared with controls, which did not meet significance (*P* = 0.0711 and *P* = 0.0581, respectively; Extended Data Fig. 3a). The patient with MIS-C who was seronegative for the SNX8 autoantibody was also negative for SNX8 autoreactive T cells.

## HLA type A*02 is more likely to present the shared epitope

MIS-C has been associated with HLA alleles A*02, B*35 and C*04 (ref. 16). The Immune Epitope Database and Analysis Resource (https://IEDB.org)[42] was used to rank the HLA class I (HLA-I) peptide presentation likelihoods for both SNX8 and SARS-CoV-2 nucleocapsid protein with respect to the MIS-C-associated HLA alleles. The distribution of predicted HLA-I-binding scores for nucleocapsid protein and SNX8 fragments matching the (ML)Q(ML)PQG SNX8/MADS motif relative to fragments lacking a match was compared. For HLA-A*02, predicted HLA-I binding was significantly higher (*P* = 8.78 × 10⁻¹⁰ for nucleocapsid protein; *P* = 0.0112 for SNX8) for fragments containing the putative autoreactive motif. There was no statistical difference for HLA-B*35 and HLA-C*04 predictions (Extended Data Fig. 3b,c). Of note, of the seven patients with MIS-C with SNX8 autoreactive T cells, at least five were positive for HLA-A*02 (Extended Data Fig. 3a). To experimentally validate HLA-I-binding predictions to SNX8 and MADS peptides, we measured peptide–HLA (pHLA) monomer stability using a β2 microglobulin (β2m) fold test, which is a proxy for pHLA-binding affinity in which anti-β2m staining reports on the strength of the pHLA complex[43]. SNX8 (MQMPQGNPL) and MADS (LQLPQGITL) peptides were loaded onto unfolded HLA-A*02:01, HLA-A*02:06 or HLA-B*35:01 monomers and stained with an anti-β2m fluorescent antibody. Consistent with the IEDB rankings, both HLA-A*02 alleles bound SNX8 and MADS peptides, with HLA-A*02:06 exhibiting the highest pHLA complex stability (Extended Data Fig. 3d).

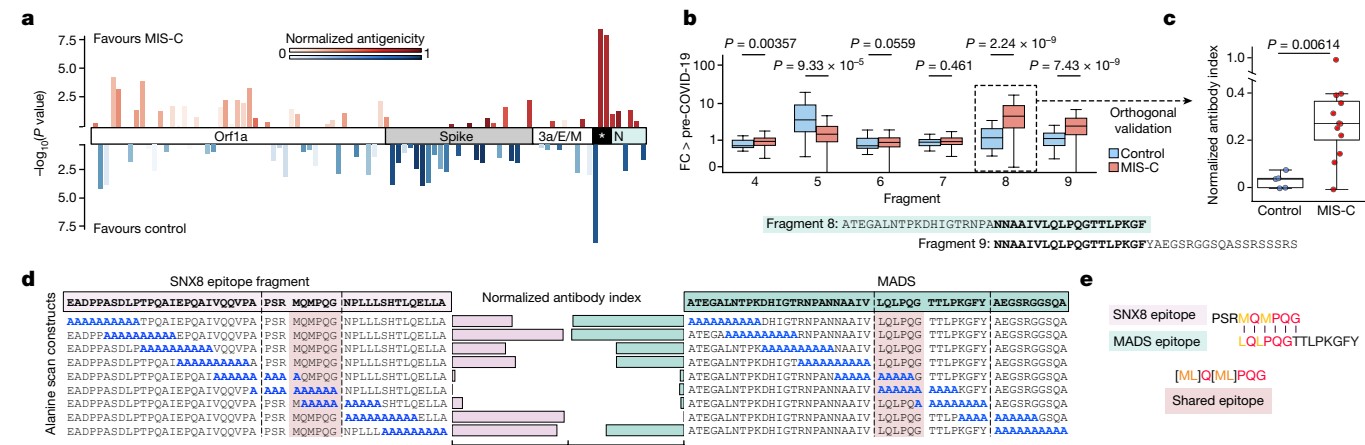

**Fig. 3 | Antibodies from patients with MIS-C preferentially target a distinct region of the SARS-CoV-2 nucleocapsid protein. a**, Relative PhIP-seq signal (FC over the mean) of 48 controls who are pre-COVID-19 (FC > pre-COVID-19) in patients with MIS-C (n = 181) and at-risk controls (n = 45) using a custom phage display library expressing the entire SARS-CoV-2 proteome to different regions of SARS-CoV-2. Only regions with a mean antibody signal of more than 1.5-fold above pre-COVID-19 controls are shown. Antigenicity (sum of the mean FC > pre-COVID-19 in MIS-C and at-risk controls) are represented by darker shades. The length of the bars represents the statistical difference in signal between MIS-C and at-risk controls to a particular region (−log₁₀ of two-sided Kolmogorov–Smirnov test P values), with upward deflections representing enrichment in MIS-C versus at-risk controls, and downward deflections representing less signal in MIS-C. The asterisk indicates the differentially reactive region of the nucleocapsid (N) protein. **b**, Bar plots showing the PhIP-seq signal (FC > pre-COVID-19) across the specific region of the

SARS-CoV-2 nucleocapsid protein (fragments 4–9) with the most divergent response in MIS-C samples (n = 181) relative to at-risk controls (n = 45), compared using a two-sided Kolmogorov–Smirnov test (exact P values are shown in the figure). The amino acid sequence of the region with the highest relative enrichment in MIS-C is highlighted in green and referred to as MADS. **c**, Strip plots and box plots showing MADS SLBA enrichments (normalized antibody indices) in patients with MIS-C (n = 11) relative to at-risk controls (n = 5). **d**, SLBA signal (normalized antibody indices) for full sequential alanine mutagenesis scans within the same three individuals for SNX8 (left) and MADS (right). Each identified epitope is bounded by black vertical dotted lines. **e**, Multiple sequence alignment of SNX8 and MADS epitopes with the amino acid sequence for the similarity region shown (for the text in colour, biochemically similar is in orange, and identical is in red). For the box plots (**b**,**c**), the whiskers extend to 1.5 times the IQR from the quartiles. The boxes represent the IQR, and the centre lines represent the median.

## T cells from patients with MIS-C are cross-reactive to the SNX8 and nucleocapsid protein similarity regions

Given the prediction that HLA types associated with MIS-C preferentially display peptides containing the similarity regions for both SNX8 and the SARS-CoV-2 nucleocapsid protein, we sought to determine whether cross-reactive T cells were present and whether they were associated with MIS-C. We stimulated peripheral blood mononuclear cells (PBMCs) from three patients with MIS-C and three at-risk controls with peptides from either the SNX8 similarity region (MQMPQGNPL) or the MADS similarity region (LQLPQGITL) for 7 days to enrich for CD8⁺ T cells reactive to these epitopes. We then built differently labelled HLA-I tetramers loaded with either the SNX8 or MADS peptides and measured binding to T cells (Extended Data Fig. 4a). We detected cross-reactive CD8⁺ T cells, which bound both peptide epitopes, in all three patients with MIS-C, whereas no cross-reactive CD8⁺ T cells were observed in at-risk controls (Extended Data Fig. 4b).

As SNX8-responsive T cells were observed in patients with MIS-C, we next asked whether the region of SNX8 similar to the SARS-CoV-2 MADS region was sufficient to activate patient T cells. A pool of 20 10-mer peptides with 9-amino acid overlaps centred on the target motif from SNX8 (collectively spanning amino acids 44–72) was used to stimulate PBMCs from two patients with MIS-C and four at-risk controls. Both patients with MIS-C had activation of T cells, whereas none of the four controls had T cell activation (Extended Data Fig. 4c).

## Identification of ex vivo cross-reactive T cell receptors

Having determined that patients with MIS-C, but not controls, contained putative SNX8/MADS cross-reactive CD8⁺ T cells, we next sought to identify T cell receptor (TCR) sequences with specificity for both the SARS-CoV-2 MADS and the host SNX8 epitopes. To do

this, PBMCs were obtained during the first 72 h of hospital admission from four study participants with HLA-A*02 and confirmed MIS-C (one individual previously identified as having putative cross-reactive T cells, and three new patients). Given that MIS-C PBMCs represent a scarce resource, we chose to expand one aliquot of PBMCs from each of the four participants (distinct from our previous peptide expansion protocol; see Methods) to maximize the chances of isolating putative cross-reactive TCRs. Although the frequency of ex vivo autoantigen-specific CD8⁺ T cells are extraordinarily low in peripheral blood, even for bona fide T cell-mediated autoimmune diseases such as T1DM[38] and multiple sclerosis[44,45], we nevertheless utilized the remaining PBMCs from each participant for direct ex vivo analysis without previous expansion. To isolate the antigen-specific TCRs, participant cells (both ex vivo and following peptide expansion) were stained using the same tetramer-labelling strategy, which previously identified the putative cross-reactive TCRs (Extended Data Fig. 4a); any cell exhibiting binding to at least two peptide-loaded tetramers was individually sorted and full-length paired TCRα and TCRβ sequences were determined. This resulted in 259 complete TCR sequences, comprising 30 and 18 unique T cell clones from the ex vivo and peptide expansion experiments, respectively. A complete list of TCR sequences is provided (Fig. 4 source data).

Next, we sought to validate the specificity of putative SNX8/MADS cross-reactive TCRs identified from the tetramer sorting, and further analyse features of the recovered TCRs. Because clusters of similar TCRs tend to recognize similar peptide antigens, a TCR similarity network was constructed from all 259 full-length TCR sequences using a previously established TCR distance metric (TCRdist)[46,47] (Fig. 4b and Extended Data Fig. 4d). In two of the four patients, we identified unique populations of clonally expanded T cells expressing putative cross-reactive TCRs directly ex vivo, whereas each of the four patients had at least one ex vivo putative cross-reactive TCR (Fig. 4b). To confirm the

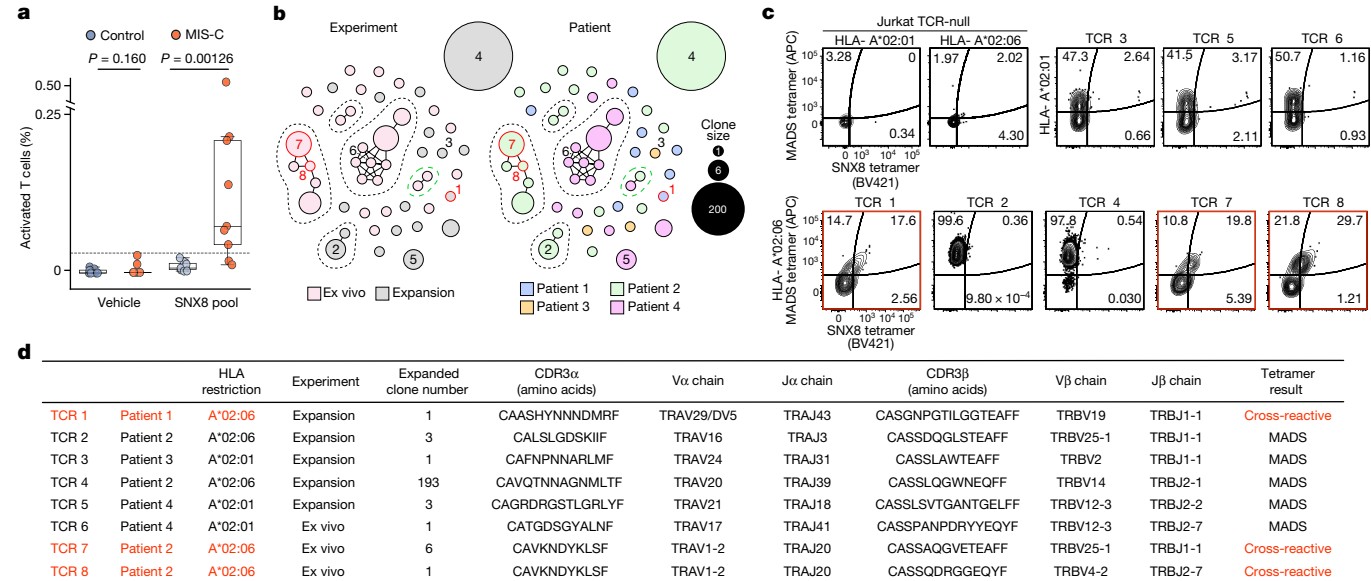

**Fig. 4 | SNX8 autoreactive CD8+ T cells in patients with MIS-C are cross-reactive to the nucleocapsid protein. a**, Strip plots and box plots showing the distribution of T cells activated in response to either vehicle (culture media + 0.2% DMSO) or the SNX8 peptide pool (SNX8 peptide + culture media + 0.2% DMSO) in patients with MIS-C (*n* = 9) and controls (*n* = 10). The relative signal was compared using a two-sided Mann–Whitney *U*-test (exact *P* values are shown in the figure). The box plot whiskers extend to 1.5 times the IQR from the quartiles, the boxes represent the IQR, and the centre lines represent the median. The dashed line is 3 s.d. above the mean of the controls in the SNX8 pool condition. **b**, TCRdist similarity network of 48 unique, paired TCRαβ sequences (*n* = 259 sequences) obtained from four patients with MIS-C. CD8+ T cells were sorted from PBMCs directly ex vivo or after 10 days of peptide expansion and staining with A*02:01 or A*02:06 HLA class I tetramers loaded with MADS (LQLPQGITL) and SNX8 (MQMPQGNPL) peptides. Each node represents a unique TCR clonotype. Edges connect nodes with a TCRdist score of less than 150. The dashed lines surround TCR similarity clusters. The node size corresponds to the T cell clone size. Nodes are coloured based on the HLA experiment type (left) or patient (right). TCRs selected for further testing are numbered TCR 1–8. The convergent node is circled in green. **c**, Specificity of putative cross-reactive TCRs expressed in Jurkat-76 cells by HLA-A*02:01 or HLA-A*02:06 tetramers loaded with MADS (LQLPQGITL) and SNX8 (MQMPQGNPL) peptides. Jurkat-76 (TCR-null) cells were used as tetramer background staining controls. The gate values indicate the frequency of MADS–APC+ and/or SNX8–BV421+ cells as the percentage of the total PE+ cells (combination staining with MADS–PE and SNX8–PE tetramers). TCRs with confirmed cross-reactivity are indicated in red. Outliers are shown. Flow plots are representative of two independent evaluations. **d**, Summary of TCR sequencing results of the eight TCRs tested.

specificity of the TCRs identified in our tetramer sorting, we selected eight TCR sequences for additional validation and generated individual cell lines that stably expressed one TCR of interest (Extended Data Fig. 5a). These Jurkat-TCR+ cell lines were tetramer stained, and cross-reactivity was confirmed in three of the Jurkat-TCR+ cell lines (TCR 1, 7 and 8; Fig. 4c). Of these validated cross-reactive TCRs, two were obtained from ex vivo PBMCs from patients with MIS-C including TCR 7, which was clonally expanded. The minimum ex vivo frequency of TCR 7 alone was more than 1 in 25,000 (6 of 140,035) circulating CD8+ T cells. The two cross-reactive TCRs obtained from the ex vivo isolation were derived from the same participant, utilize the same TRAV gene (*TRAV1-2*) with identical CDR3α sequences and clustered with three additional sequences in the TCRdist space, one of which was also clonally expanded, suggesting that this patient had an active expansion of a large cluster of SNX8/MADS cross-reactive CD8+ T cells (Fig. 4d). Furthermore, we note a cluster of two similar TCRs obtained from ex vivo sampling of different participants (patients 2 and 4) with different HLA types ('convergent node'; circled in green in Fig. 4b). Although these putative cross-reactive TCRs were not evaluated further, the cluster suggests that TCR specificities to these epitopes may converge across individuals.

The remaining five Jurkat-TCR+ cell lines (TCR 2–6) exhibited single specificity to the MADS tetramer with four of five coming from the peptide expansion. To evaluate possible interference between tetramers, which can arise when pHLA–TCR-binding affinities differ, Jurkat-TCR+ cell lines were stained with individual tetramers. The results confirm that four of these TCRs are indeed reactive only to MADS (Extended Data Fig. 5b). However, TCR 2, although showing strong binding preference to MADS, also bound the individual SNX8 tetramer, suggesting that the

higher affinity for MADS may outcompete binding to the SNX8 tetramer in some cases. This observation is in line with the notion that autoreactive cross-reactive TCRs with lower relative affinities to autoantigens may escape thymic negative selection. Finally, because the original tetramer experiments were based on an early 2020 SARS-CoV-2 minor variant sequence (LQLPQGITL), all eight Jurkat-TCR+ cell lines were also stained with HLA tetramers loaded with the SARS-CoV-2 Wuhan MADS sequence (LQLPQGTTL). In all cases, the Jurkat-TCR+ cells bound the Wuhan MADS tetramer, consistent with the notion that T cells encoding these and other similar TCRs may be capable of responding to multiple SARS-CoV-2 strains (Extended Data Fig. 5c).

## RNA expression profile of *SNX8* during SARS-CoV-2 infection

As previously discussed, *SNX8* is expressed across multiple tissues, but is highest in immune cells, consistent with its role in defending against RNA viruses via recruitment of MAVS[27]. To further investigate the potential impact of combined B cell and T cell autoimmunity to SNX8 following SARS-CoV-2 infection, we used scRNA-seq to analyse *SNX8* expression in PBMCs from patients with severe, mild or asymptomatic SARS-CoV-2 infection or influenza infection and uninfected healthy controls[48]. Following SARS-CoV-2 infection, *SNX8* had the highest mean expression in classical and non-classical monocytes and B cells (Extended Data Fig. 6a,b) and was elevated in individuals infected with SARS-CoV-2 compared with those who were uninfected (Extended Data Fig. 6c). Within myeloid lineage cells, *SNX8* expression correlated with *MAVS* expression and *OAS1* and *OAS2* (which encode two known regulators of the MAVS pathway implicated in MIS-C pathogenesis[7])

expression (Extended Data Fig. 6d). Conversely, *SNX8* expression is inversely correlated to SARS-CoV-2 infection severity. This follows a similar pattern to *OAS1* and *OAS2*. However, unlike *OAS1*, *OAS2* and *MAVS*, *SNX8* is preferentially expressed during SARS-CoV-2 infection compared with influenza virus infection (Extended Data Fig. 6e).

## Discussion

The SARS-CoV-2 pandemic largely spared children from severe disease. One rare but notable exception is MIS-C, an enigmatic and life-threatening syndrome. Previous studies have surfaced numerous associations, but have failed to identify a direct mechanistic link between SARS-CoV-2 and MIS-C. In this study, 199 samples from patients with MIS-C and 45 paediatric at-risk controls were analysed using customized human and SARS-CoV-2 proteome PhIP-seq libraries. Targeted follow-up experiments from these assays ultimately revealed that patients with MIS-C preferentially had antibodies targeting the epitope motif (ML)Q(ML)PQG shared by both the SARS-CoV-2 nucleocapsid protein and the human protein SNX8. Cross-reactive CD8[+] T cells targeting both regions were detected in patients with MIS-C, but not in controls, suggesting that these CD8[+] T cells may contribute to immune dysregulation through the inappropriate targeting of immune cells expressing SNX8. We found evidence that the (ML)Q(ML)PQG epitope motif elicits both B cell and T cell reactivity; further study of this epitope convergence is warranted.

These findings help to connect several important known aspects of MIS-C pathophysiology and draw parallels to other diseases in which exposure to a new antigen leads to autoimmunity, such as paraneoplastic autoimmune disease or cross-reactive epitopes between Epstein–Barr virus and host proteins in multiple sclerosis[17–19,22,26]. An expansion of T cells expressing TCRβ variable gene 11-2 (*TRBV11-2*) has been shown in MIS-C[8,15,16]; however, the underlying driver remains unknown. Although we did not observe an overrepresentation of *TRBV11-2* in our putative cross-reactive TCR dataset, we did identify two expanded TRBV11-2[+] clones (*n* = 6 and *n* = 2) sequenced directly from ex vivo samples. Although SNX8 is a relatively understudied protein, it has been linked to the function and activity of MAVS[27]. Dysregulation of the MAVS antiviral pathway, by inborn errors of immunity, has been shown to underlie certain cases of MIS-C[7]. The most straightforward connection linking MIS-C to SNX8 may be through an inappropriate autoimmune response against tissues with elevated MAVS pathway expression. These results are the first to directly link the initial SARS-CoV-2 infection and the subsequent development of MIS-C. We propose that MIS-C may be the result of multiple uncommon events converging. The initial insult is probably the formation of a combined B cell and T cell response that preferentially targets a particular motif within the MADS region of the SARS-CoV-2 nucleocapsid protein. In a subset of individuals, these B cell and T cell responses cross-react to the self-protein SNX8. This cross-reactive motif has strong binding characteristics for the MIS-C-associated HLA-A*02 (ref. 16), further indicating that this may be an important risk factor in the development of MIS-C.

Using conservative criteria (3 s.d. greater than controls by targeted immunoprecipitation of the epitope-containing peptide), at least 17% of sera from patients with MIS-C are autoreactive for SNX8; however, approximately 37% of sera from patients with MIS-C yielded detectable enrichment compared with controls in the entire dataset. Because we only tested for a single epitope target, we are unable to determine the upper limits of the in vivo frequency of cross-reactive CD8[+] T cells in patients with MIS-C. Our results suggest that the frequency of these cross-reactive CD8[+] T cells is within the range of 1 in 10,000–100,000 CD8[+] T cells. This substantially exceeds the frequency of antigen-specific autoreactive CD8[+] T cells found in peripheral circulation in bona fide T cell-mediated autoimmune diseases such as T1DM[38] and multiple sclerosis[44,45]. Similar to T1DM, the autoreactive and cross-reactive CD8[+] T cells in patients with MIS-C may be found at

far greater abundance within peripheral tissues known to be affected by the disease[38]. Even accounting for these limitations, our results describe a subset of MIS-C, indicating that other mechanisms probably exist. Antibodies to ERFL are present in many children with MIS-C who do not have autoreactivity to SNX8, and *ERFL* has a highly similar tissue RNA expression profile as *SNX8* (second-most similar among all known proteins; Human Protein Atlas)[36]. If autoreactive T cells to ERFL indeed exist, they would be predicted to engage a nearly identical set of cells and tissues. It is important to also consider that MIS-C prevalence has rapidly decreased as an increasing number of children have developed immunity through vaccination and natural SARS-CoV-2 infection. We speculate that perhaps this could be related to the strong deviation of the anti-SARS-CoV-2 immune response away from the critical MADS region of the nucleocapsid protein that we have identified, to other major epitopes such as those in the spike protein through vaccination and past infection[49]. Supporting this notion is recent CDC surveillance, which noted that more than 80% (92 of 112) of individuals with MIS-C in 2023 were in unvaccinated children (but vaccine eligible), and that the majority of children who developed MIS-C despite previous vaccination probably had waned immunity[50].

MIS-C is complex, and more work will be required to fully understand this syndrome. The results of this study, and specifically the development of combined cross-reactive B cells and T cells, build on other notable examples of molecular mimicry; however, the mechanisms by which the presence of a cross-reactive epitope forces a break in tolerance remain unclear. Our results shed light on how one post-infectious disease (MIS-C) develops, yielding insights that may help better explain, diagnose and ultimately treat a range of additional conditions associated with infections.

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

**Overcoming COVID-19 Network Investigators**

**Laura L. Loftis**[18], **Charlotte V. Hobbs**[19], **Keiko M. Tarquinio**[20], **Michele Kong**[21], **Julie C. Fitzgerald**[22], **Paula S. Espinal**[23], **Tracie C. Walker**[24], **Stephanie P. Schwartz**[24], **Hillary Crandall**[25], **Katherine Irby**[26], **Mary Allen Staat**[27], **Courtney M. Rowan**[28], **Jennifer E. Schuster**[29], **Natasha B. Halasa**[30], **Shira J. Gertz**[31], **Elizabeth H. Mack**[32], **Aline B. Maddux**[33], **Natalie Z. Cvijanovich**[34] & **Matt S. Zinter**[1]

[18]Department of Pediatrics, Division of Critical Care Medicine, Baylor College of Medicine, Houston, TX, USA. [19]Department of Pediatrics, Division of Infectious Diseases, University of Mississippi Medical Center, Jackson, MS, USA. [20]Department of Pediatrics, Division of Critical Care Medicine, Emory University School of Medicine, Children's Healthcare of Atlanta, Atlanta, GA, USA. [21]Department of Pediatrics, Division of Pediatric Critical Care Medicine, University of Alabama at Birmingham, Birmingham, AL, USA. [22]Department of Anesthesiology and Critical Care, Children's Hospital of Philadelphia, University of Pennsylvania, Perelman School of Medicine, Philadelphia, PA, USA. [23]Personalized Medicine and Health Outcomes Research, Nicklaus Children's Hospital, Miami, FL, USA. [24]Department of Pediatrics, University of North Carolina at Chapel Hill Children's Hospital, Chapel Hill, NC, USA. [25]Department of Pediatrics, Division of Pediatric Critical Care, University of Utah, Primary Children's Hospital, Salt Lake City, UT, USA. [26]Section of Pediatric Critical Care, Department of Pediatrics, Arkansas Children's Hospital, Little Rock, AR, USA. [27]Department of Pediatrics, Division of Infectious Diseases, University of Cincinnati and Cincinnati Children's Hospital Medical Center, Cincinnati, OH, USA. [28]Department of Pediatrics, Division of Pediatric Critical Care Medicine, Indiana University School of Medicine and Riley Hospital for Children, Indianapolis, IN, USA. [29]Department of Pediatrics, Division of Pediatric Infectious Diseases, Children's Mercy Kansas City, Kansas City, MO, USA. [30]Department of Pediatrics, Division of Pediatric Infectious Diseases, Vanderbilt University Medical Center, Nashville, TN, USA. [31]Department of Pediatrics, Division of Pediatric Critical Care, Cooperman Barnabas Medical Center, Livingston, NJ, USA. [32]Division of Pediatric Critical Care Medicine, Medical University of South Carolina, Charleston, SC, USA. [33]Department of Pediatrics, Section of Critical Care Medicine, University of Colorado School of Medicine and Children's Hospital Colorado, Aurora, CO, USA. [34]Division of Critical Care Medicine, UCSF Benioff Children's Hospital Oakland, Oakland, CA, USA.

## Methods

### Patients

Patients were recruited through the prospectively enrolling multicentre Overcoming COVID-19 and Taking on COVID-19 Together study in the USA. All patients meeting clinical criteria were included in the study, and therefore no statistical methods were used to predetermine sample size and no blinding or randomization of subjects occurred. The study was approved by the central Boston Children's Hospital Institutional Review Board (IRB) and reviewed by IRBs of participating sites with CDC IRB reliance. A total of 292 patients consented and were enrolled into one of the following independent cohorts between 1 June 2020 and 9 September 2021: 223 patients hospitalized with MIS-C (199 in the primary discovery cohort and 24 in a separate subsequent validation cohort), 29 patients hospitalized for COVID-19 in either an intensive care or step-down unit (referred to as 'severe acute COVID-19' in this study) and 45 outpatients (referred to as 'at-risk controls' in this study) post-SARS-CoV-2 infections associated with mild or no symptoms. The demographic and clinical data are summarized in Extended Data Tables 1–3. The 2020 US CDC case definition was used to define MIS-C[51]. All patients with MIS-C had positive SARS-CoV-2 serology results and/ or positive SARS-CoV-2 test results by reverse transcriptase quantitative PCR. All patients with severe COVID-19 or outpatient SARS-CoV-2 infections had a positive antigen test or nucleic acid amplification test for SARS-CoV-2. For outpatients, samples were collected from 36 to 190 days after the positive test (median of 70 days after a positive test; interquartile range of 56–81 days). For use as controls in the SARS-CoV-2-specific PhIP-seq, plasma from 48 healthy, pre-COVID-19 controls were obtained as deidentified samples from the New York Blood Center. These samples were part of retention tubes collected at the time of blood donations from volunteer donors who provided informed consent for their samples to be used for research.

### DNA oligomers for SLBAs

DNA coding for the desired peptides for use in SLBAs were inserted into split luciferase constructs containing a terminal HiBiT tag and synthesized (Twist Biosciences) as DNA oligomers and verified by Twist Biosciences before shipment. Constructs were amplified by PCR using the 5′- AAGCAGAGCTCGTTTAGTGAACCGTCAGA-3′ and 5′-GGCCGGCCGTTTAAACGCTGATCTT-3′ primer pair.

For SNX8, the oligomers coded for the following sequences:
EADPPASDLPTPQAIEPQAIVQQVPAPSRMQMPQGNPLLLSHTLQELLA
AAAAAAAAAAATPQAIEPQAIVQQVPAPSRMQMPQGNPLLLSHTLQ
ELLA
EADPPAAAAAAAAAAAEPQAIVQQVPAPSRMQMPQGNPLLLSHTLQ
ELLA
EADPPASDLPAAAAAAAAAAAVQQVPAPSRMQMPQGNPLLLSHTLQ
ELLA
EADPPASDLPTPQAIAAAAAAAAAAAPSRMQMPQGNPLLLSHTLQ
ELLA
EADPPASDLPTPQAIEPQAIAAAAAAAAAAAQMPQGNPLLLSHTLQELLA
EADPPASDLPTPQAIEPQAIVQQVPAAAAAAAAAAANPLLLSHTLQELLA
EADPPASDLPTPQAIEPQAIVQQVPAPSRMAAAAAAAAAAASHTLQELLA
EADPPASDLPTPQAIEPQAIVQQVPAPSRMQMPQGAAAAAAAAAA
ELLA
EADPPASDLPTPQAIEPQAIVQQVPAPSRMQMPQGNPLLLAAAAA
AAAA

For SARS-CoV-2 nucleocapsid protein, the oligomers coded for the following sequences:
ATEGALNTPKDHIGTRNPANNAAIVLQLPQGTTLPKGFYAEGSRGGSQA
AAAAAAAAAAADHIGTRNPANNAAIVLQLPQGTTLPKGFYAEGSRG
GSQA
ATEGAAAAAAAAAAAARNPANNAAIVLQLPQGTTLPKGFYAEGSRG
GSQA
ATEGALNTPKAAAAAAAAAAANAAIVLQLPQGTTLPKGFYAEGSRGGSQA

ATEGALNTPKDHIGTAAAAAAAAAAALQLPQGTTLPKGFYAEGSRG
GSQA
ATEGALNTPKDHIGTRNPANAAAAAAAAAAAGTTLPKGFYAEGSRG
GSQA
ATEGALNTPKDHIGTRNPANNAAIVAAAAAAAAAAAKGFYAEGSRG
GSQA
ATEGALNTPKDHIGTRNPANNAAIVLQLPQAAAAAAAAAAAEGSRG
GSQA
ATEGALNTPKDHIGTRNPANNAAIVLQLPQGTTLPAAAAAAAAAAAGSQA
ATEGALNTPKDHIGTRNPANNAAIVLQLPQGTTLPKGFYAAAAAAAAAA

### DNA plasmids for RLBAs

For RLBAs, DNA expression plasmids under control of a T7 promoter and with a terminal Myc–DDK tag for the desired protein were utilized. For ERFL, a custom plasmid was ordered from Twist Bioscience in which a Myc–DDK-tagged full-length *ERFL* sequence under a T7 promoter was inserted into the pTwist Kan High Copy Vector (Twist Bioscience). Twist Bioscience verified a sequence-perfect clone by next-generation sequencing before shipment. Upon receipt, the plasmid was sequence verified by Primordium Labs. For SNX8, a plasmid containing the Myc–DDK-tagged full-length human *SNX8* under a T7 promoter was ordered from Origene (RC205847) and was sequence verified by Primordium Labs upon receipt. For KDELR1, a plasmid containing the Myc–DDK-tagged full-length human *KDELR1* under a T7 promoter was ordered from Origene (RC205880) and was sequence verified by Primordium Labs upon receipt. For IL1RN, a plasmid containing the Myc–DDK-tagged full-length human *IL1RN* under a T7 promoter was ordered from Origene (RC218518) and was sequence verified by Primordium Labs upon receipt.

### Polypeptide pools for activation-induced marker assays

To obtain polypeptides tiling the full-length SNX8 protein, 15-mer polypeptide fragments with 11-amino acid overlaps were ordered from JPT Peptide Technologies and synthesized. Together, a pool of 130 of these polypeptides (referred to as the 'SNX8 pool') spanned all known translated SNX8 (the full-length 465-amino acid SNX8 protein, as well as a unique region of SNX8 isoform 3). A separate pool was designed to cover primarily the region of SNX8 with similarity to the SARS-CoV-2 nucleocapsid protein in high resolution (referred to as the 'high-resolution epitope pool'). This pool contained 20 10-mers with 9-amino acid overlaps tiling amino acids 44–72 (IVQQVPAPSRMQMPQGNPLLLSHTLQELL) of the full-length SNX8 protein. The sequence of each of these 150 polypeptides was verified by mass spectrometry and purity was calculated by high-performance liquid chromatography (HPLC).

### Peptides for tetramer assays

For use in loading tetramers, three peptides were ordered from Genemed Synthesis as 9-mers. LQLPQGTTL and LQLPQGITL correspond to the region of the SARS-CoV-2 nucleocapsid protein with similarity to human SNX8 in the ancestral sequence and a minor variant, respectively. This sequence was verified by mass spectrometry and purity was calculated as 96.61% by HPLC. The other sequence, MQMPQGNPL, corresponds to the region of human SNX8 protein with similarity to the SARS-CoV-2 nucleocapsid protein. This sequence was verified by mass spectrometry and purity was calculated as 95.83% by HPLC.

### Human proteome PhIP-seq

Human proteome PhIP-seq was performed following our previously published vacuum-based PhIP-seq protocol[12] (https://www.protocols.io/view/scaled-high-throughput-vacuum-phip-protocol-ewov1459kvr2/v1).

Our human peptidome library consists of a custom-designed phage library of 731,724 unique T7 bacteriophage each presenting a different 49-amino acid peptide on its surface. Collectively, these peptides tile the entire human proteome including all known isoforms (as of 2016) with 25-amino acid overlaps. Of the phage library, 1 ml was incubated

with 1 μl of human serum overnight at 4 °C and immunoprecipitated with 25 μl of 1:1 mixed protein A and protein G magnetic beads (10008D and 10009D, Thermo Fisher). These beads were than washed, and the remaining phage–antibody complexes were eluted in 1 ml of *Escherichia coli* (BLT5403, EMD Millipore) at 0.5–0.7 OD and amplified by growing in a 37 °C incubator. This new phage library was then re-incubated with the serum from the same individual and the previously described protocol was repeated. DNA was then extracted from the final phage library, barcoded, PCR amplified and Illumina adaptors were added. Next-generation sequencing was performed using an Illumina sequencer (Illumina) to a read depth of approximately 1 million per sample.

### Human proteome PhIP-seq analysis
All human peptidome analysis (except when specifically stated otherwise) was performed at the gene level, in which all reads for all peptides mapping to the same gene were summed, and 0.5 reads were added to each gene to allow inclusion of genes with zero reads in mathematical analyses. Within each individual sample, reads were normalized by converting to the percentage of total reads. To normalize each sample against background nonspecific binding, a fold change over mock-IP was calculated by dividing the sample read percentage for each gene by the mean read percentage of the same gene for the AG bead-only controls. This fold-change signal was then used for side-by-side comparison between samples and cohorts. Fold-change values were also used to calculate $z$ scores for each patient with MIS-C compared with controls and for each control sample by using all remaining controls. These $z$ scores were used for the logistic-regression feature weighting. In instances of peptide-level analysis, raw reads were normalized by calculating the number of reads per 100,000 reads.

### SARS-CoV-2 proteome PhIP-seq
SARS-CoV-2 proteome PhIP-seq was performed as previously described[39]. In brief, 38 amino acid fragments tiling all open reading frames from SARS-CoV-2, SARS-CoV-1 and 7 other CoVs were expressed on T7 bacteriophage with 19-amino acid overlaps. Of the phage library, 1 ml was incubated with 1 μl of human serum overnight at 4 °C and immunoprecipitated with 25 μl of 1:1 mixed protein A and protein G magnetic beads (10008D and 10009D, Thermo Fisher). Beads were washed five times on a magnetic plate using a P1000 multichannel pipette. The remaining phage–antibody complexes were eluted in 1 ml of *E. coli* (BLT5403, EMD Millipore) at 0.5–0.7 OD and amplified by growing in 37 °C incubator. This new phage library was then re-incubated with the serum of the same individual and the previously described protocol was repeated for a total of three rounds of immunoprecipitations. DNA was then extracted from the final phage library, barcoded, PCR amplified and Illumina adaptors were added. Next-generation sequencing was then performed using an Illumina sequencer (Illumina) to a read depth of approximately 1 million per sample.

### Coronavirus proteome PhIP-seq analysis
To account for differing read depths between samples, the total number of reads for each peptide fragment was converted to the number of reads per 100,000 (RPK). To calculate normalized enrichment relative to pre-COVID-19 controls (FC > pre-COVID-19), the RPK for each peptide fragment within each sample was divided by the mean RPK of each peptide fragment among all pre-COVID-19 controls. These FC > pre-COVID-19 values were used for all subsequent analyses as described in the text and figures.

### RLBA
RLBAs were performed as previously described[12,32]. In brief, DNA plasmids containing full-length cDNA under the control of a T7 promoter for each of the validated antigens (see 'DNA plasmids for RLBAs' above) were verified by Primordium Labs sequencing. The respective DNA templates were used in the T7 TNT in vitro transcription/translation

kit (L1170, Promega) using [35S]-methionine (NEG709A, PerkinElmer). Respective protein was column purified on Nap-5 columns (17-0853-01, GE Healthcare), and equal amounts of protein (approximately 35,000 counts per minute) were incubated overnight at 4 °C with 2.5 μl of serum or 1 μl of anti-Myc-positive control antibody (1:10 dilution; 2272S, Cell Signaling Technology). Immunoprecipitation was then performed on 25 μl of Sephadex protein A/G beads (4:1 ratio; GE17-5280-02 and GE17-0618-05, Sigma-Aldrich) in 96-well polyvinylidene difluoride filtration plates (EK-680860, Corning). After thoroughly washing, the counts per minute of immunoprecipitated protein was quantified using a 96-well Microbeta Trilux liquid scintillation plate reader (Perkin Elmer).

### SLBA
SLBA was performed as previously described[52]. A detailed SLBA protocol is available on protocols.io (https://doi.org/10.17504/protocols.io.4r3l27b9pg1y/v1).

In brief, the DNA oligomers listed above (see 'DNA oligomers for SLBAs') were amplified by PCR using the primer pairs listed above (see 'DNA oligomers for SLBAs'). Unpurified PCR product was used as input in the T7 TNT in vitro transcription/translation kit (L1170, Promega) and the Nano-Glo HiBit Lytic Detection System (N3040, Promega) was used to measure relative luciferase units of translated peptides in a luminometre. Equal amounts of protein (in the range of $2 \times 10^6$–$2 \times 10^7$ relative luciferase units) were incubated overnight with 2.5 μl patient sera or 1 μl anti-HiBit-positive control antibody (1:10 dilution; CS2006A01, Promega) at 4 °C. Immunoprecipitation was then performed on 25 μl of Sephadex protein A/G beads (1:1 ratio; GE17-5280-02 and GE17-0618-05, Sigma-Aldrich) in 96-well polyvinylidene difluoride filtration plates (EK-680860, Corning). After thoroughly washing, luminescence was measured using the Nano-Glo HiBit Lytic Detection System (N3040, Promega) in a luminometre.

### Activation-induced marker assay
PBMCs were obtained from ten patients with MIS-C and ten controls for use in the activation-induced marker assay. PBMCs were thawed, washed, resuspended in serum-free RPMI medium and plated at a concentration of $1 \times 10^6$ cells per well in a 96-well round-bottom plate. For each individual, PBMCs were stimulated for 24 h with either the SNX8 pool (see above) at a final concentration of 1 mg ml$^{-1}$ per peptide in 0.2% DMSO or a vehicle control containing 0.2% DMSO only. For four of the controls and two of the patients with MIS-C, there were sufficient PBMCs for an additional stimulation condition using the SNX8 high-resolution epitope pool (see above) also at a concentration of 1 mg ml$^{-1}$ per peptide in 0.2% DMSO for 24 h. Following the stimulation, cells were washed with FACS buffer (Dulbecco's PBS without calcium or magnesium, 0.1% sodium azide, 2 mM EDTA and 1% FBS) and stained with the following antibody panel each at 1:100 dilution for 20 min at 4 °C, and then flow cytometry analysis was immediately performed.

For the antibody panel: anti-CD3 Alexa 647 (clone OKT3, 317312, BioLegend), anti-CD4 Alexa 488 (clone OKT4, 317420, BioLegend), anti-CD8 Alexa 700 (clone SK1, 344724, BioLegend), anti-OX-40 (also known as CD134) PE-Dazzle 594 (clone ACT35, 350020, BioLegend), anti-CD69 PE (clone FN-50, 310906, BioLegend), anti-CD137 (also known as 4-1BB) BV421 (clone 4B4-1, 309820, BioLegend), anti-CD14 PerCP-Cy5 (clone HCD14, 325622, BioLegend), anti-CD16 PerCP-Cy5 (clone B73.1, 360712, BioLegend), anti-CD19 PerCP-Cy5 (clone HIB19, 302230, BioLegend) and Live/Dead Dye eFluor 506 (65-0866-14, Invitrogen).

The activation-induced marker analysis was performed using FlowJo software using the gating strategy shown in Extended Data Fig. 7a. All gates were fixed within each condition of each sample. Activated CD4 T cells were defined as those that were co-positive for OX40 and CD137. Activated CD8 T cells were defined as those that were co-positive for CD69 and CD137. Gating thresholds for activation were defined by the outer limits of signal in the vehicle controls allowing for up to two outlier cells. Frequencies were calculated as a percentage of total CD3$^+$

cells (T cells). Two MIS-C samples had insufficient total events captured by flow cytometry (total of 5,099 and 4,919 events, respectively) and were therefore removed from analysis.

## Initial tetramer assay

For the initial tetramer assay, see Extended Data Fig. 4a. PBMCs from two patients with MIS-C with HLA-A*02:01 (HLA typed from PAXgene RNAseq, one confirmed by serotyping), one patient with MIS-C with HLA-B*35:01 (HLA typed from PAXgene RNAseq) and three at-risk controls with HLA-A*02.01 (all three identified by serotyping, two of three confirmed by PAXgene RNAseq HLA typing; the other sample did not have genomic DNA available for genotyping) were thawed, washed and put into culture with media containing recombinant human IL-2 at 10 ng ml$^{-1}$ in 96-well plates. The peptide fragments (details above) LQLPQGITL and MQMPQGNPL were then added to PBMCs to a final concentration of 10 mg ml$^{-1}$ per peptide and incubated (37 °C at 5% CO$_2$) for 7 days.

Following the 7 days of incubation, a total of eight pHLA class I tetramers were generated from UV-photolabile biotinylated monomers, four each from HLA-A*02:01 and HLA-B*35:01 (NIH Tetramer Core). Peptides were loaded via UV peptide exchange. Tetramerization was carried out using streptavidin conjugated to fluorophores PE and APC or BV421 followed by quenching with 500 μM D-biotin, similar to our previously published methods[44,53]. Tetramers were then pooled together as shown below:

For the HLA-A*02:01 pool, the MADS (LQLPQGITL)-loaded PE tetramer, MADS (LQLPQGITL)-loaded APC tetramer, SNX8 (MQMPQGNPL)-loaded PE tetramer and SNX8 (MQMPQGNPL)-loaded BV421 tetramer were used, all with HLA-A*02:01 restriction.

For the HLA-B*35:01 pool, the MADS (LQLPQGITL)-loaded PE tetramer, MADS (LQLPQGITL)-loaded APC tetramer, SNX8 (MQMPQGNPL)-loaded PE tetramer and SNX8 (MQMPQGNPL)-loaded BV421 tetramer were used, all with HLA-B*35:01 restriction.

All PBMCs were then treated with 100 nM dasatinib (StemCell) for 30 min at 37 °C followed by staining (no wash step) with the respective tetramer pool corresponding to their HLA restriction (final concentration of 2–3 μg ml$^{-1}$) for 30 min at 25 °C. Cells were then stained with the following cell-surface markers each at 1:100 dilution for 20 min, followed by immediate analysis on a flow cytometer.

For the surface markers: anti-CD8 Alexa 700 (clone SK1, 357404, BioLegend), anti-CD4 PerCP-Cy5 (clone SK1, 300530, BioLegend), anti-CD14 PerCP-Cy5 (clone HCD14, 325622, BioLegend), anti-CD16 PerCP-Cy5 (clone B73.1, 360712, BioLegend), anti-CD19 PerCP-Cy5 (clone HIB19, 302230, BioLegend) and Live/Dead Dye eFluor 506 (65-0866-14, Invitrogen). Streptavidin was conjugated to PE (S866, Invitrogen), APC (S868, Invitrogen) and BV421 (405225, BioLegend).

The gating strategy is outlined in Extended Data Fig. 7b. A stringent tetramer gating strategy was used to identify cross-reactive T cells, in which CD8$^+$ T cells were required to be triple positive for PE, APC and BV421 labels (that is, a single CD8 T cell bound to PE-conjugated LQLPQGITL and/or PE-conjugated MQMPQGNPL in addition to APC-conjugated LQLPQGITL and BV421-conjugated MQMPQGNPL).

Serotyping was performed using an anti-HLA-A2 antibody (1:100 dilution; FITC anti-human HLA-A2 antibody, clone BB7.2, 343303, BioLegend), and pertinent results are shown in Extended Data Fig. 7c.

## Assembly of easYmer monomers and fold testing

For the assembly of HLA class I pHLA easYmer monomers and fold testing, see Fig. 4. Unfolded, biotinylated easYmer monomers (Immudex) were obtained for HLA-A*02:01 and HLA-A*02:06. SARS-CoV-2 MADS (LQLPQGITL), SARS-CoV-2 Wuhan (LQLPQGTTL) and human SNX8 (MQMPQGNPL) peptides were commercially synthesized (Genscript), diluted to 1 mM in ddH$_2$O or DMSO, and loaded onto each easYmer allele according to the manufacturer's instructions at 18 °C for 48 h. Proper pHLA monomer formation and MADS and SNX8 peptide-binding

strength were evaluated for each HLA using a 'β2m fold test' relative to negative (no peptide; unloaded monomer) and positive (strong binding peptide; CMV pp65 495–503 (NLVPMVATV)) controls as per the manufacturer's protocol. In brief, peptide-loaded monomers with a concentration of 500 nM were serially diluted to 9 nM, 3 nM and 1 nM in dilution buffer (1× PBS with 5% glycerol; G5516, Sigma-Aldrich) and incubated with streptavidin beads (6–8 μm; SVP-60-5, Spherotech) at 37 °C for 1 h to allow binding of stable complexes to beads, then washed three times with FACS buffer (1× PBS, 0.5% BSA (A7030, Sigma-Aldrich) and 2 mM EDTA (15575-038, Thermo Fisher Scientific)). Samples were then stained with PE-conjugated anti-human β2m antibody (clone BBM.1, sc-13565, Santa Cruz Biotech) at 1:200 for 30 min at 4 °C, washed three times with FACS buffer and analysed on a 5 Laser 16UV-16V-14B-10YG-8R AURORA spectral cytometer (Cytek). pHLA-binding strength positively correlated with stability and concentration of the pHLA–β2m complex. Therefore, the geometric mean fluorescence intensity of anti-β2m staining in this assay reports on the strength of the pHLA binding compared with the positive and negative controls. We classified binding strength for each HLA and peptide combination based on the fold change in anti-β2m geometric mean fluorescence intensity over the no-peptide negative control at 9 nM. Strong binders were defined at more than 10-fold higher, moderate binders at more than 3-fold, weak binders at more than 1.5-fold and non-binders at less than 1.5-fold change over the negative control. Flow cytometry data were analysed using FlowJo version 10.7.2 software (BD Biosciences).

## pHLA tetramer assembly

For the pHLA tetramer assembly, see Fig. 4. pHLA tetramers were assembled from HLA-A*02:01 and HLA-A*02:06 easYmer monomers (Immudex) with confirmed peptide binding to SARS-CoV-2 MADS (LQLPQGITL), Wuhan (LQLPQGTTL) and SNX8 (MQMPQGNPL) peptides according to the manufacturer's instructions. In brief, fluorochrome-conjugated streptavidin (0.2 mg ml$^{-1}$, PE, 405203, BioLegend; 0.2 mg ml$^{-1}$, APC, 405207, BioLegend; and BV421, 405226, BioLegend) was added to loaded monomers at 8 ng per 1 μl pHLA complex (500 nM) in three volumes. After each 1/3 volume addition, samples were mixed and incubated for 15 min at 4 °C in the dark. Assembled tetramers were stored at 4 °C in the dark until use.

## Enhanced peptide-specific T cell expansion

For enhanced peptide-specific T cell expansion, see Fig. 4. PBMCs from MIS-C confirmed participants with HLA-A*02:01 or HLA-A*02:06 were obtained for peptide-specific expansion according to published methods[54] before single-cell sorting of tetramer-positive T cells. On expansion day 0, PBMCs were thawed, counted and seeded onto 96-well round-bottom plates at 100,000 cells per well in 200 μl antigen-presenting cell differentiation media (X-VIVO 15 serum-free haematopoietic cell medium (04-418Q, Lonza) supplemented with human GM-CSF (1,000 IU ml$^{-1}$; 130-095-372, Miltenyi Biotec), human IL-4 (500 IU ml$^{-1}$; 204-IL-010, R&D Systems) and human Flt3-L (50 ng ml$^{-1}$; 308-FKN-025, R&D Systems) final concentrations) and incubated for 24 h at 37 °C and 5% CO$_2$. On day 1, 100 μl cell supernatant was replaced with 100 μl Adjuvant Solution (X-VIVO 15 supplemented with R848 (10 μM; tlrl-r848-5, InvivoGen), lipopolysaccharide (*Salmonella minnesota*; 100 ng ml$^{-1}$; tlrl-smlps, InvivoGen) and human IL-1β (10 ng ml$^{-1}$; 201-LB-010, R&D Systems) final concentrations) and pooled MADS (LQLPQGITL) and SNX8 (MQMPQGNPL) peptides at a final concentration of 10 μM each. No-peptide control wells were set up for each sample by adding a 1:2 dilution of DMSO in H$_2$O to match the peptide volume and diluent. Cells were incubated for 24 h at 37 °C and 5% CO$_2$. On days 2, 4, 7 and 9, 100 μl supernatant was replaced with 100 μl T cell expansion solution: RP-10 (RPMI 1640 (22400-089, Gibco), 10% heat-inactivated human serum AB (100-512, Gemini Bio-Products), 10 mM HEPES, 0.1 mg ml$^{-1}$ gentamicin (15750-060, Thermo Fisher Scientific) and 1× GlutaMAX (35050-061, Gibco)) supplemented with human

IL-2 (10 IU ml$^{-1}$; 202-IL-050, R&D Systems), human IL-7 (10 ng ml$^{-1}$; 207-IL-025, R&D Systems) and human IL-15 (10 ng ml$^{-1}$; 200-15, PeproTech) final concentrations. On day 10, peptide-expanded cells from an individual participant were pooled; cells from no-peptide controls were collected separately.

### Single-cell index sorting

Unexpanded PBMCs (direct ex vivo) or peptide-expanded T cells were obtained, washed in 1× PBS and treated with 100 nM dasatinib (CDS023389, Sigma-Aldrich) in 1× PBS for 30 min at 37 °C and 5% CO$_2$ (ref. 55). Cells were then pelleted and resuspended in 50 µl FACS buffer (1× PBS and 0.04% BSA) supplemented with human TruStain FcX blocking buffer (1:10 dilution; 422302, BioLegend), 500 µM D-biotin (B20656, Thermo Fisher Scientific) and a unique tetramer cocktail containing MADS−tetramer−PE (1:10 dilution), MADS−tetramer−APC (1:10 dilution), SNX8−tetramer−PE (1:10 dilution) and SNX8−tetramer−BV421 (1:10 dilution) based on participant HLA type (A*02:01 and A*02:06). Cells were incubated in the dark at 25 °C for 1 h followed by direct addition of 50 µl (100 µl total volume) of FACS supplemented with 500 µM D-biotin and an antibody cocktail containing FITC-conjugated anti-human CD3 (1:20 dilution; clone OKT3, lot B390808, 317306, BioLegend), BV605-conjugated anti-human CD8 (1:20 dilution; clone SK1, lot B371925, 344742, BioLegend), BV510-conjugated anti-human CD4 (1:20 dilution; clone OKT4, lot B375526, 317444, BioLegend), BV510-conjugated anti-human CD14 (1:20 dilution; clone 63D3, lot B390770, 367124, BioLegend), BV510-conjugated anti-human CD16 (1:20 dilution; clone 3G8, lot B372132, 302048, BioLegend), BV510-conjugated anti-human CD19 (1:20 dilution; clone HIB19, lot B390665, 302242, BioLegend) and Ghost Dye Violet 510 Viability Dye (1:400 dilution; lot D0870061322133, 13-0870-T500, Tonbo Biosciences) for 30 min in the dark at 4 °C. Cells were then pelleted, washed twice with 4 ml FACS buffer (containing 500 µM D-biotin), suspended in 500 µl FACS (containing 500 µM D-biotin) and passed through a 45-µM filter before proceeding to single-cell sorting on a Sony SY3200 cell sorter. Individual, live, BV510 dump gate (CD4, CD14, CD16 and CD19)-negative, CD3$^+$CD8$^+$ T lymphocytes were gated to distinguish tetramer triple-positive cells (PE$^+$APC$^+$BV421$^+$) as described in Extended Data Fig. 7d and sorted into individual wells of a 384-well plate loaded with Superscript VILO master mix (11754250, Thermo Fisher Scientific). After sorting, plates were centrifuged at 500$g$ and stored at −80 °C until processing.

### Paired TCRαβ amplification and sequencing

Single-cell paired TCRα and TCRβ chain library preparation and sequencing was performed on T cells sorted into 384-well index plates as previously described[56]. In brief, after reverse transcription of cells sorted in Superscript VILO master mix, cDNA underwent two rounds of nested multiplex PCR amplification using a mix of human V-segment-specific forward primers and human TRAC and TRBC segment-specific reverse primers (see Supplementary Table 1 for primer details). Resulting TCRα and TCRβ amplicons were sequenced on an Illumina MiSeq at 2 × 150-bp read length.

### Cell lines

All cultured cell lines were maintained at 37 °C and 5% CO$_2$ in a humidified incubator. HEK 293T cells (CRL-3216, American Type Culture Collection) were purchased from the American Type Culture Collection and verified commercially. HEK 293T cells were cultured in DMEM (11965-092, Gibco) supplemented with 10% FBS (16140-071, Gibco), 2 mM L-glutamine (25030-081, Gibco) and 100 U ml$^{-1}$ penicillin–streptomycin (15140-122, Gibco). 2D3 Jurkat J76.7 cells[57,58] (TCR-null, CD8$^+$) expressing an NFAT−eGFP reporter were kindly provided by F. Fujiki and were cultured in RPMI 1640 (22400-089, Gibco) supplemented with 10% FBS, 2 mM L-glutamine and 100 U ml$^{-1}$ penicillin–streptomycin. All cell lines were confirmed to be mycoplasma negative during the course of experiments.

### TCR repertoire analysis

TCR similarity networks were constructed as previously described[49,59]. In brief, to measure the distance between TCRαβ clonotypes, we used the TCRdist algorithm implementation from the CoNGA v0.1.2 Python package[47]. Further analysis was performed using the R language for statistical computing, with merging and subsetting of data performed using the dplyr v1.1.4 package. TCR similarity networks were built using stringdist v0.9.12 and igraph v2.0.3 (ref. 60) R packages, and visualized using gephi v0.9.7 (ref. 61) software.

### TCR reconstruction and cloning

Full-length TCRαβ sequences were reconstructed from V/J gene usage and CDR3 sequences using Stitchr v1.0.0 (ref. 62) for each index-sorted T cell. TCRα and TCRβ chain sequences were modified to use murine constant regions and joined by a 2A element from thosea asigna virus (T2A). A sequence encoding mCherry was additionally appended by a 2A element from porcine teschovirus (P2A) as a fluorescent marker of transduction. The full-length gene fragment encoding TCRβ−T2A−TCRα−P2A−mCherry was synthesized and cloned commercially (Genscript) into the lentiviral vector pLVX-EF1α-IRES-Puro (631253, Takara).

### Generation of TCR-expressing Jurkat cells

To generate transducing particles packaging individual TCRs of interest (Fig. 4c), HEK 293T cells were transduced with a pLVX lentiviral vector encoding a unique TCRαβ−mCherry insert, psPAX2 packaging plasmid (plasmid #12260, Addgene) and an pMD2.G envelope plasmid (plasmid #12259, Addgene) at a ratio of 4:3:1. At 24 h and 48 h post-transfection, viral supernatants were harvested, passed through a 0.45-µm SFCA filter (723-9945, Thermo Fisher Scientific), concentrated using Lenti-X Concentrator (631232, Takara) and stored at −80 °C as single-use aliquots. To generate TCR-expressing Jurkat cell lines (Jurkat-TCR$^+$), 2D3 Jurkat J76.7 cells (TCR-null, CD8$^+$, NFAT−eGFP reporter) were seeded in a 12-well tissue-culture-treated plate at $1 \times 10^6$ cells per well in complete RPMI (RPMI 1640, 10% FBS, 2 mM L-glutamine, 100 U ml$^{-1}$ penicillin−streptomycin) and transduced by adding concentrated lentivirus dropwise to each well. At 48–72 h post-tranduction, puromycin was added at 1 µg ml$^{-1}$ and cultured for 1 week to select for transduced cells. Jurkat-TCR$^+$ cell lines were validated for the presence of correctly folded TCR on the cell surface by flow cytometry using a monoclonal antibody targeting the mouse TCRβ constant region (APC/Fire750-conjugated; clone H57-597, 109246, BioLegend; Extended Data Fig. 5a). Flow cytometry data were collected on a custom-configured BD Fortessa using FACSDiva software (v8.0.1; Becton Dickinson) and analysed using FlowJo version 10.7.2 software (BD Biosciences).

### Specificity validation of putative cross-reactive TCR sequences

The specificity of TCR-expressing Jurkat T cell lines was validated by tetramer staining using the same reagents used for single-cell sorting PBMCs (above). In brief, $1 \times 10^6$ Jurkat-TCR$^+$ cell lines or untransduced Jurkat J76.7 (TCR-null; background control) were washed in 1× PBS and resuspended in 50 µl FACS buffer (1× PBS and 0.04% BSA) and a unique tetramer cocktail containing MADS−tetramer−PE (1:10 dilution), MADS−tetramer−APC (1:10 dilution), SNX8−tetramer−PE (1:10 dilution) and SNX8−tetramer−BV421 (1:10 dilution) based on the restricting HLA type (A*02:01 and A*02:06). Tetramers conjugated to the Wuhan peptide sequence (LQLPQGTTL), including Wuhan−tetramer−PE (1:10 dilution) and Wuhan−tetramer−APC (1:10 dilution), were also tested. A second set of wells were set up in which each individual tetramer was used to stain cells. Cells were incubated in the dark at 25 °C for 30 min after which 50 µl of FACS buffer containing Ghost Dye Violet 510 Viability Dye (1:400 dilution; lot D0870061322133, 13-0870-T500, Tonbo Biosciences) was added for an additional 30-min incubation in the dark at 25 °C. Cells were then washed twice with 1 ml FACS buffer, suspended in 300 µl FACS and analysed by flow cytometry

on a custom-configured BD Fortessa using FACSDiva software (v8.0.1; Becton Dickinson). Cell population gating and fluorescence analysis was performed using FlowJo version 10.7.2 software (BD Biosciences) as described in Extended Data Fig. 7e.

## scRNA-seq analysis

To assess the cell-type specificity in a relevant disease context, we analysed *SNX8* expression from a single-cell sequencing of PBMC samples from patients with severe, mild or asymptomatic COVID-19 infection, influenza virus infection and healthy controls[48]. Gene expression data from 59,572 pre-filtered cells were downloaded from the Gene Expression Omnibus database under accession GSE149689 for analysis and downstream processing with scanpy v1.10.0 (ref. 63). Cells with (1) less than 1,000 total counts, (2) less than 800 expressed genes, and (3) more than 3,000 expressed genes were filtered out as further quality control, leaving 42,904 cells for downstream analysis. Gene expression data were normalized to have 10,000 counts per cell and were log1p transformed. Highly variable genes were calculated using the scanpy function highly_variable_genes using Seurat flavor with the default parameters (min_mean = 0.0125, max_mean = 3, and min_disp = 0.5)[64]. Only highly variable genes were used for further analysis. The total number of counts per cell was regressed out, and the gene expression matrix was scaled using the scanpy function scale with max_value = 10. Dimensionality reduction was performed using principal components analysis with 50 principal components. Batch balanced $k$-nearest neighbours, implemented with scanpy's function bbknn, was used to compute the top neighbours and normalize batch effects[65]. The batch-corrected cells were clustered using the Leiden algorithm and projected into two dimensions with uniform manifold approximation and projection for visualization. Initial cluster identity was determined by finding marker genes with differential expression analysis performed using a Student's $t$-test on log1p-transformed raw counts with the scanpy function rank_genes_groups[66,67].

## Statistical methods

All statistical analysis was performed in Python using the Scipy Stats package unless otherwise indicated. For comparisons of distributions of PhIP-seq enrichment between two groups, a non-parametric Kolmogorov–Smirnov test was utilized. For logistic-regression feature weighting, the Scikit-learn package[68] was used, and logistic-regression classifiers were applied to $z$-scored PhIP-seq values from individuals with MIS-C versus at-risk controls. A liblinear solver was used with L1 regularization, and the model was evaluated using a five-fold cross-validation (four of the five for training, and one of the five for testing). For the RLBAs and SLBAs, first an antibody index was calculated as follows: (sample value − mean blank value)/(positive control antibody values − mean blank values). For the alanine mutagenesis scans, blank values of each construct were combined, and a single mean was calculated. A normalization function was then applied to the experimental samples only (excluding antibody-only controls) to create a normalized antibody index ranging from 0 to 1. Comparisons between two groups of samples were performed using a Mann–Whitney $U$-test. An antibody was considered to be 'positive' when the normalized antibody index in a sample was greater than 3 s.d. above the mean of controls. When comparing two groups of normally distributed data, a Student's $t$-test was performed.

## Reporting summary

Further information on research design is available in the Nature Portfolio Reporting Summary linked to this article.

## Data availability

The published article includes all datasets generated or analysed as a part of this study. Individual source data are provided with

associated figures (where appropriate) per the data-sharing agreement stipulated under the Ruth L. Kirschstein National Research Service Award Individual Postdoctoral Fellowship (award no. F32AI157296 to R.C.M.). Raw flow cytometry source files can be made available on reasonable request. All PhIP-seq data are publicly available via a Dryad digital repository (https://doi.org/10.7272/Q6SJ1HVH). Raw TCR reads are available through the NCBI Sequence Read Archive (SRA) BioProject PRJNA1110271, with associated BioSample accession numbers SAMN41334731, SAMN41334732, SAMN41334730 and SAMN41334729. Source data are provided with this paper.

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

**Acknowledgements** The following members of the Overcoming COVID-19 Network Investigators study group were all closely involved with the design, implementation and oversight of the Overcoming COVID-19 study, as well as collecting patient samples and data: M. Kong, H. Kelley, M. Murdock, C. Colston. K. V. Typpo, K. Irby, R. C. Sanders Jr, M. Yates, C. Smith, N. Z. Cvijanovich, M. S. Zinter, A. B. Maddux, E. Port, R. Mansour, S. Shankman, N. Baig, F. Zorensky, P. S. Espinal, B. Chatani, G. McLaughlin, K. M. Tarquinio, K. Jones, B. M. Coates, C. M. Rowan, A. G. Randolph, M. M. Newhams, S. Kucukak, T. Novak, E. R. McNamara, H. Kyung Moon, T. Kobayashi, J. Melo, S. R. Jackson, M. K. Echon Rosales, C. Young, S. R. Chen, J. Chou, R. Da Costa Aguiar, M. Gutierrez-Arcelus and M. Elkins. The Taking On COVID-19 Together team include: D. Williams, L. Williams, L. Cheng, Y. Zhang, D. Crethers, D. Morley, S. Steltz, K. Zakar, M. A. Armant, F. Ciuculescu, H. R. Flori, M. K. Dahmer, E. R. Levy, S. Behl, N. M. Drapeau, C. V. Hobbs, J. E. Schuster, A. Kietzman, S. Hill, M. L. Cullimore, R. J. McCulloh, S. J. Gertz, S. P. Schwartz, T. C. Walker, R. A. Nofziger, M. A. Staat, C. C. Rohlfs, J. C. Fitzgerald, R. Burnett, J. Bush, E. H. Mack, N. Reed, N. B. Halasa, L. L. Loftis, H. Crandall and K. K. Ampofo. Members of the US Centers for Disease Control and Prevention COVID-19 Response Team on the Overcoming COVID-19 study were L. D. Zambrano, M. M. Patel and A. P. Campbell. The authors acknowledge the New York Blood Center for contributing pre-COVID-19 healthy donor blood samples, which were used as controls for the SARS-CoV-2 library PhIP-seq. The authors acknowledge the contributions of W. Browne and S. Pleasure for their work investigating potential central nervous system-specific autoimmunity in MIS-C; T. Kharel for help designing the Python code used in the analysis; D. Blauvelt for ideas regarding the application of advanced statistics to PhIP-seq data analysis; and S. A. Schattgen for thoughtful discussion of TCR sequencing and TCR similarity network analysis and help with deposition of the TCR sequencing into the Sequence Read Archive. BioRender (https://biorender.com) was used to build graphics for Fig. 1a and Extended Data Fig. 4a. This work was supported by the Pediatric Scientist Development Program and the Eunice Kennedy Shriver National Institute of Child Health and Human Development (K12-HD000850 to A.B.), and the Chan Zuckerberg Biohub SF (to J.L.D. and M.S.A.). Overcoming COVID-19 Study Network enrolment, patient data and specimen collections were supported by the CDC contracts 75D30120C07725, 75D30121C10297

and 75D30122C13330 from the Centers for Disease Control and Prevention to Boston Children's Hospital to A.G.R. and the National Institute of Allergy and Infectious Diseases (R01AI154470) to A.G.R. Patient clinical data and specimens also collected at Boston Children's Hospital for the Taking On COVID-19 Together (TOCT) study were supported in part by the Boston Children's Hospital Emerging Pathogens and Epidemic Response Cluster of Clinical Research Excellence and the Institutional Centers for Clinical and Translational Research to A.G.R. and K.L.M. P.G.T. is supported by the American Lebanese Syrian Associated Charities at St. Jude Children's Research Hospital (SJCRH) and funding from the National Institute of Allergy and Infectious Diseases (5R01AI154470-03, 2R01AI136514-06, 3P01AI165077-01S1, 75N93021C00016 and U01 AI144616). R.C.M. is supported by a Ruth L. Kirschstein National Research Service Award Individual Postdoctoral Fellowship award (F32AI157296).

**Author contributions** A.B., R.C.M., J.J.S.Jr, S.E.V., J.C., P.G.T., A.G.R., M.S.A. and J.L.D. conceptualized the study. A.B., R.C.M., J.J.S.Jr, S.E.V., E.R., C.R.Z., A.F.K., J.V.R., J.C., P.G.T., A.G.R., M.S.A. and J.L.D. curated the methodology. A.B., R.C.M., J.J.S.Jr, S.E.V., A.M., C.-Y.W., A.S., J.V.P., D.J.L.Y., H.K., W.A., A.M.K. and C.M.-B. performed or contributed to experiments. A.B., R.C.M., J.J.S.Jr, H.S.M., A.F.K., J.A. and M.V.P. performed the formal analysis. K.Z., T.N., L.D.Z., A.P.C., A.G.R., K.L.M. and the Overcoming COVID-19 Network Investigators acquired the patient sample and clinical data. T.N., A.G.R. and the Overcoming COVID-19 Network Investigators curated the clinical data. A.B., H.S.M. and J.L.D. wrote the original draft of the manuscript. A.B., R.C.M., J.C., T.N., H.S.M., L.D.Z., A.P.C., P.G.T., A.G.R., M.R.W., M.S.A. and J.L.D. reviewed and edited the manuscript. J.C., P.G.T., A.G.R., M.S.A. and J.L.D. supervised the study.

**Competing interests** J.L.D. reports being a founder and paid consultant for Delve Bio, Inc., and a paid consultant for the Public Health Company and Allen & Co. M.A.S. receives unrelated research funding from the National Institutes of Health, the Centers for Disease Control and Prevention, Cepheid and Merck and unrelated honoria from UpToDate, Inc. M.R.W. receives unrelated research grant funding from Roche/Genentech and Novartis, and received speaking honoraria from Genentech, Takeda, WebMD and Novartis. J.C. reports consulting fees from GLG group, payments from Elsevier for work as an Associate Editor, a patent pending for methods and compositions for treating and preventing T cell-driven diseases, payments related to participation on a Data Safety Monitoring Board or Advisory Board for Enzyvant, and is a member of the Diagnostic Laboratory Immunology Committee of the Clinical Immunology Society. M.S.Z. receives unrelated funding from the National Heart, Lung, and Blood Institute and consults for Sobi. N.B.H. reports unrelated previous grant support from Sanofi and Quidel, and current grant support from Merck. C.V.H. reports being a speaker for Biofire and a reviewer for UpToDate, Inc. and Dynamed.com. A.G.R. receives royalties as a section editor for Pediatric Critical Care Medicine UpToDate, Inc., and received honoraria for MIS-C-related Grand Round Presentations. A.G.R. is also on the medical advisor board of Families Fighting Flu and is Chair of the International Sepsis Forum, which is supported by industry and has received reagents from Illumina, Inc. P.G.T. is on the Scientific Advisory Board of Immunoscape and Shennon Bio, has received research support and personal fees from Elevate Bio, and consulted for 10X Genomics, Illumina, Pfizer, Cytoagents, Merck and JNJ. The findings and conclusions in this report are those of the authors and do not necessarily represent the views of the Centers for Disease Control and Prevention nor the National Institute of Allergy and Infectious Diseases. All other authors declare no competing interests.

**Additional information**

**Correspondence and requests for materials** should be addressed to Mark S. Anderson or Joseph L. DeRisi.

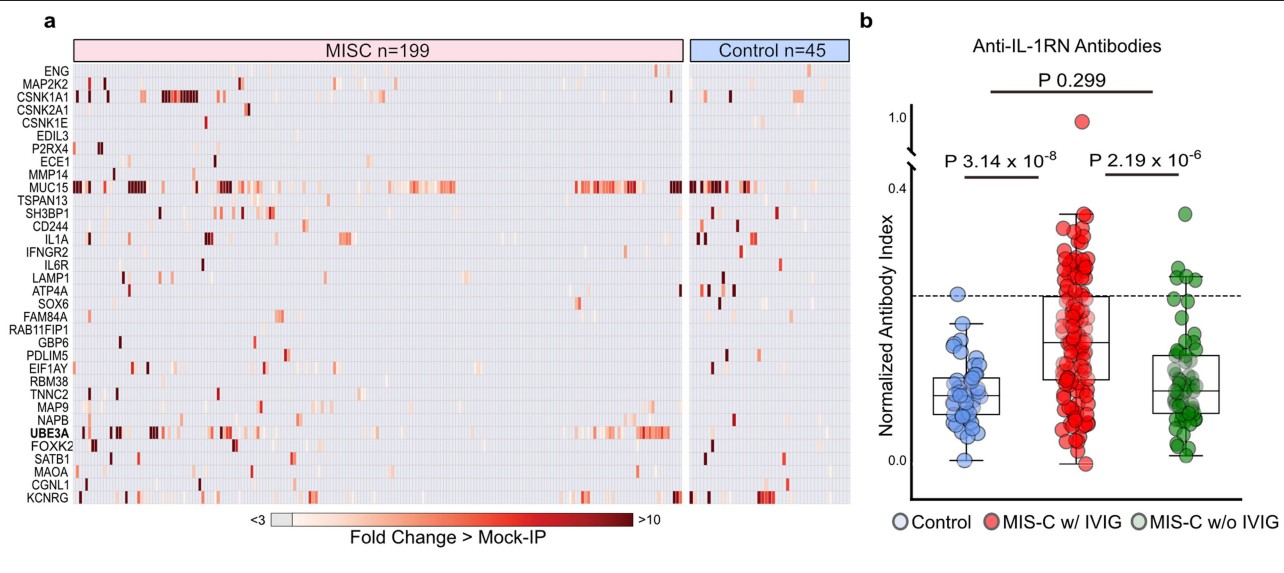

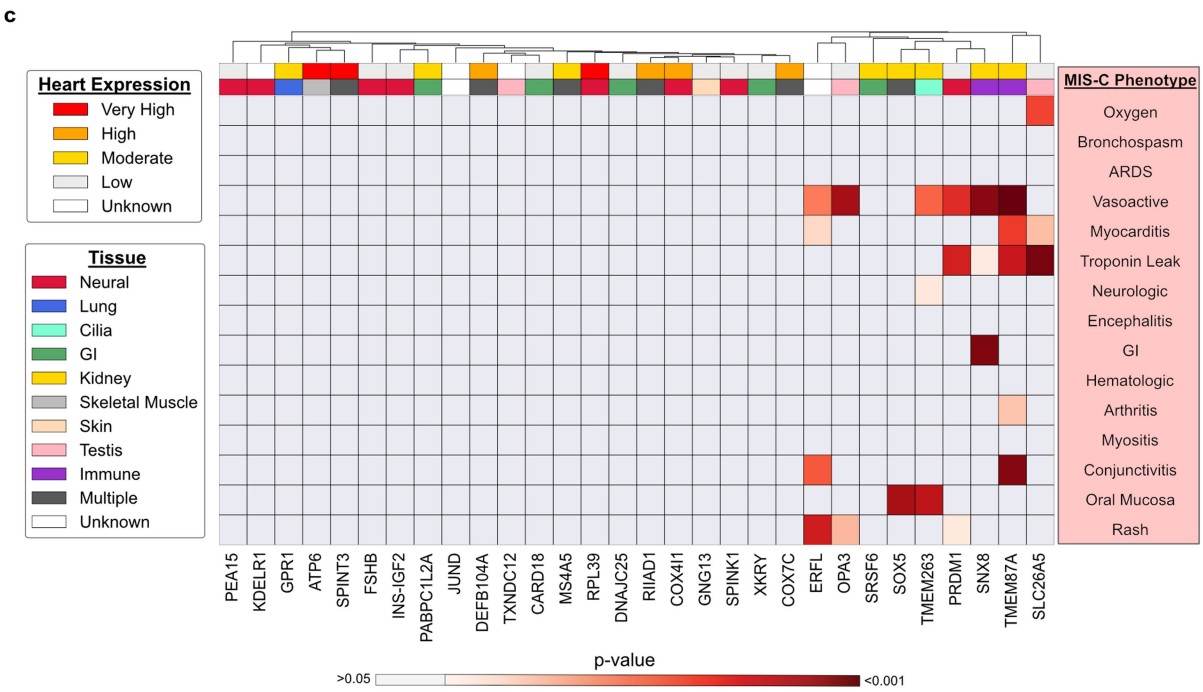

**Extended Data Fig. 1 | Previously reported autoantigens and phenotypic associations of novel autoantigens. a**, Heatmap showing distribution of PhIP-Seq enrichments (FC > Mock-IP) of previously reported MIS-C autoantibodies in MIS-C patients (*n* = 199) and at-risk controls (n = 45). (b) Stripplots and boxplots showing distribution of signal (normalized antibody index) for antibodies targeting IL-1 receptor antagonist (IL-1Ra) measured by RLBA in at-risk controls (blue; *n* = 45), MIS-C patient samples containing IVIG (red; *n* = 135), and MIS-C patient samples without IVIG (green; *n* = 61). Dotted line at 3 standard deviations above the mean of controls. Two-sided Mann-Whitney U testing was performed (exact *P* values shown in figure). **c**, Heatmap of *P* values (two-sided Kolmogorov-Smirnov testing) for differences

in autoantibody enrichment for MIS-C patients (*n* = 199) with versus without each clinical phenotype (numbers vary for each phenotype and are shown in Extended Data Table 2). Significant *P* values in the negative direction (in which there is increased signal in individuals without the phenotype) are masked (colored as *P* > 0.05). For each autoantigen, tissue RNA-sequencing data from Human Protein Atlas (Proteinatlas.org) is shown. Amount of expression in cardiac tissue in top row (Very high = nTPM >1000, High=nTPM 100-1000, Moderate=nTPM 10-100, Low=nTPM <10), and predominant tissue type in second-from-top row. Explanations of criteria for MIS-C phenotypes, and distribution of each phenotype within the cohort, can be found in Extended Data Table 2.

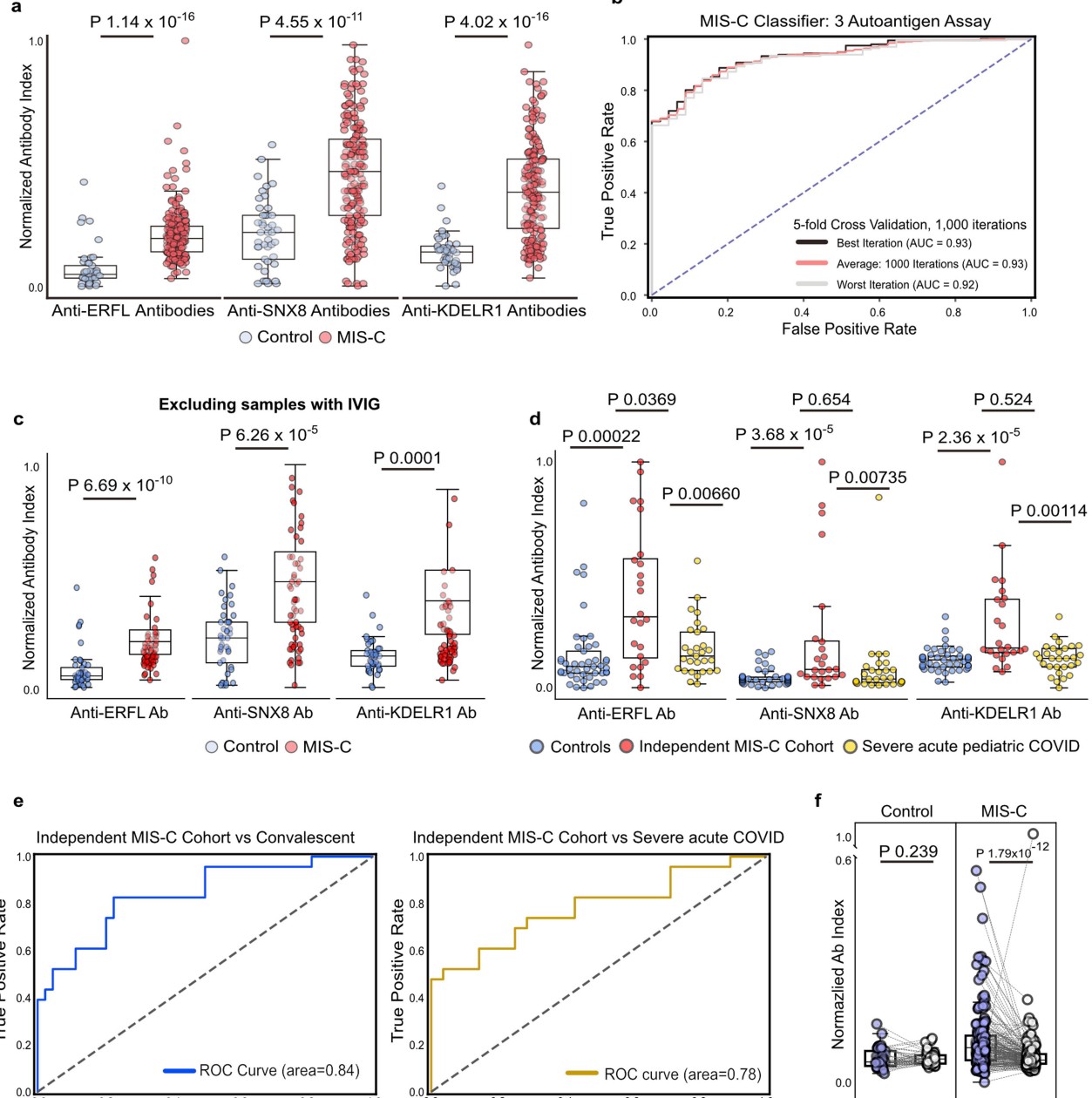

**Extended Data Fig. 2 | Orthogonally validated autoantibodies classify MIS-C and can be epitope specific. a**, Stripplots and boxplots showing radioligand binding assay (RLBA) values (normalized antibody indices) for each of the top 3 autoantibodies identified by PhIP-Seq logistic regression in individuals with MIS-C ($n = 197$ for ERFL, $n = 196$ for SNX8, n = 196 for KDELR1) and each at-risk control ($n = 45$ for ERFL, SNX8, and KDELR1). Two-sided Mann-Whitney U testing performed (exact $P$ values shown in figure). **b**, Logistic regression receiver operating characteristic (ROC) curve using RLBA values as input to distinguish MIS-C patients ($n = 196$) from at-risk controls ($n = 45$) iterated 1,000 times. **c**, Stripplots and boxplots showing RLBA enrichments (normalized antibody indices) only in those MIS-C samples without IVIG ($n = 61$ for ERFL, $n = 60$ for SNX8, $n = 60$ for KDELR1) relative to at-risk controls ($n = 45$ for ERFL, SNX8, and KDELR1). Two-sided Mann-Whitney U testing performed (exact $P$ values shown in figure). **d**, Stripplots abd boxplots showing RLBA enrichments (normalized antibody indices) for ERFL, SNX8, and KDELR1 in an independent cohort of children with MIS-C (red; $n = 24$ for each RLBA) compared to children severely ill

with acute COVID-19 (yellow; $n = 29$ for each RLBA) and at-risk controls (blue; $n = 45$ for each RLBA). Two-sided Mann-Whitney U testing performed (exact $P$ values shown in figure). **e**, Logistic regression ROC curves for classification of the independent MIS-C cohort ($n = 24$) versus at-risk controls ($n = 45$) (left) and the independent MIS-C cohort ($n = 24$) versus children severely ill with acute COVID-19 ($n = 29$) (right). **f**, Paired stripplots and boxplots showing SLBA enrichments (normalized antibody indices) in MIS-C patients ($n = 182$) and at-risk controls ($n = 45$) for the full 49 amino acid SNX8 wild-type (WT) polypeptide fragment (lavender) relative to the same SNX8 fragment with alanine mutagenesis of the [PSRMQMPQG] epitope (white). SNX8 WT fragment SLBA values are the means of technical replicates, SNX8 epitope mutagenesis values are from a single experiment. Two-sided Mann-Whitney U testing performed (exact $P$ values shown in figure). For all boxplots in the figure, the whiskers extend to 1.5 times the interquartile range (IQR) from the quartiles, the boxes represent the IQR, and centre lines represent the median.

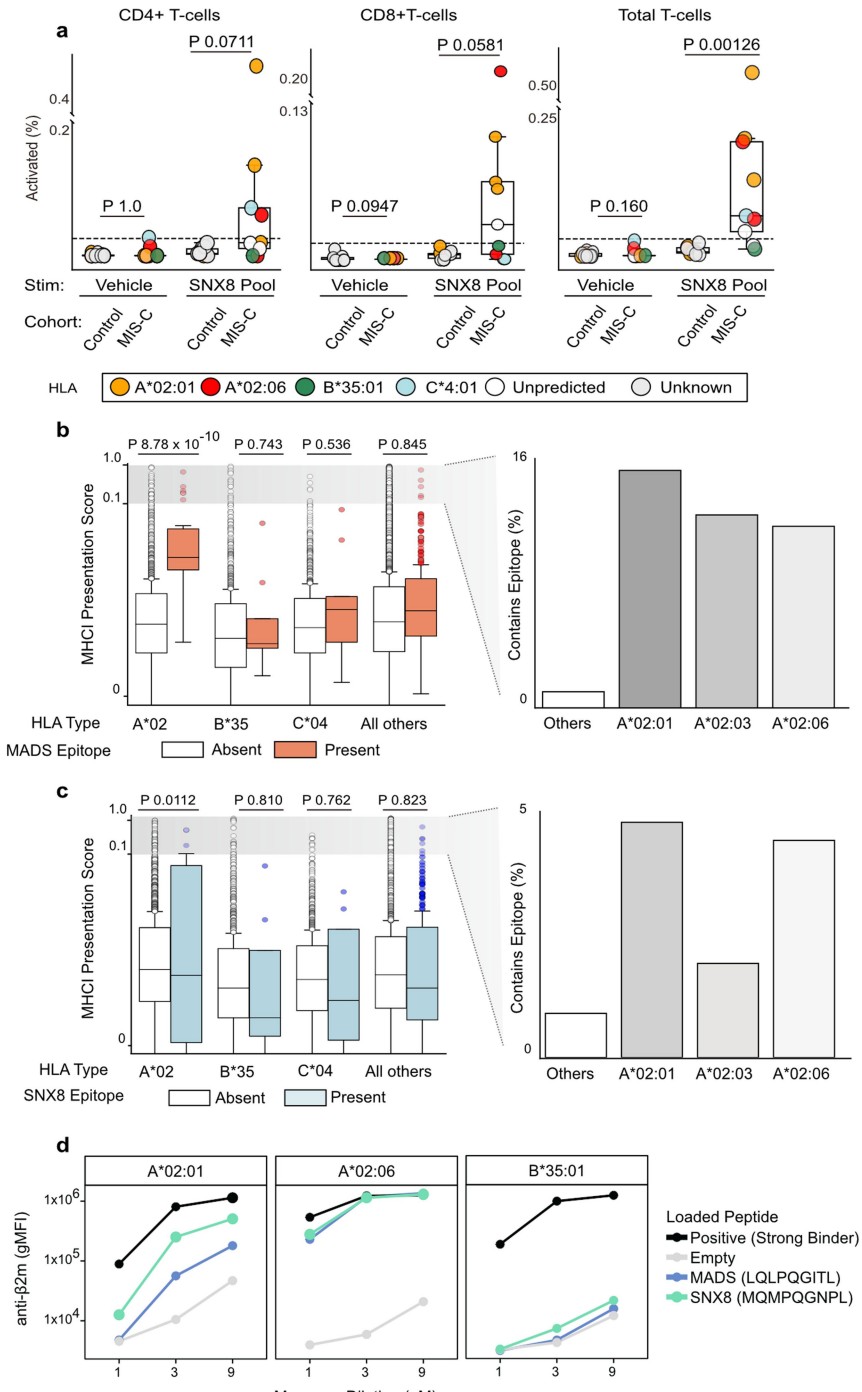

**Extended Data Fig. 3 | HLA associations of SNX8 activated T cells and HLA binding characteristics of peptides containing SNX8/MADS shared epitope motif. a**, Stripplots and boxplots showing distribution of CD4+, CD8+, and total T cells which activate in response to either vehicle (culture media + 0.2% DMSO) or SNX8 peptide pool (SNX8 peptide + culture media + 0.2% DMSO) using AIM assay in MIS-C patients (n = 9) and controls (n = 10). Patient HLA type indicated by color of dot. HLA unpredicted means patient contained none of the MIS-C associated HLA types. Dotted line at 3 standard deviations above the mean of the SNX8 stimulated controls. Two-sided Mann-Whitney U testing was performed (exact P values shown in figure). **b**, Computationally predicted HLA class I presentation scores (Immune Epitope Database; IEDB.org) for each possible peptide fragment of full-length SARS-CoV-2 N protein for each of the three MIS-C associated HLA types (A*02, B*35 and C*04) relative to a reference set of HLA-types encompassing over 99% of humans. Those fragments containing the MADS similarity region "LQLPQG" in orange. Data normally distributed; two-sided

t-tests were performed (exact P values shown in figure). Percent of fragments within each specific HLA type with a score greater than 0.1 (likely to be presented) shown on right. **c**, Identical analysis but using full length SNX8 protein rather than SARS-CoV-2 NP, and the SNX8 similarity region "MGMPQG" rather than the MADS region "LQLPQG". Data normally distributed; two-sided t-tests were performed (exact P values shown in figure). **d**, HLA binding results from β2m folding assay for SARS-CoV-2 N and SNX8 peptides representative of two independent evaluations. Each peptide tested for binding in HLA-A*02:01, A*02:06, and B*35:01 class I monomers. Data presented as geometric mean fluorescence intensity (gMFI) of PE-conjugated anti-human β2m antibody staining of peptide-HLA monomers relative to negative (no peptide; unloaded HLA monomer) and positive (strong binding peptide; CMV pp65 495-503 [NLVPMVATV]) controls. For all boxplots in the figure, the whiskers extend to 1.5 times the interquartile range (IQR) from the quartiles, the boxes represent the IQR, and centre lines represent the median.

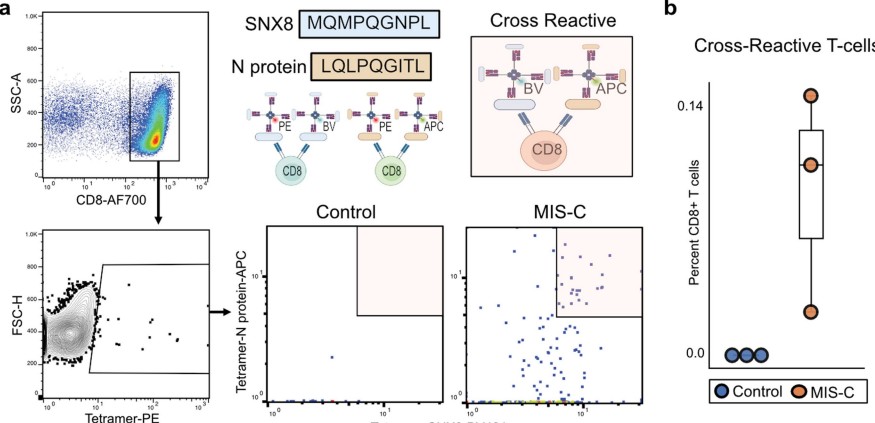

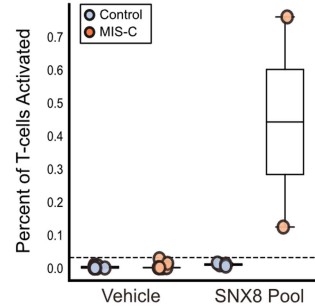

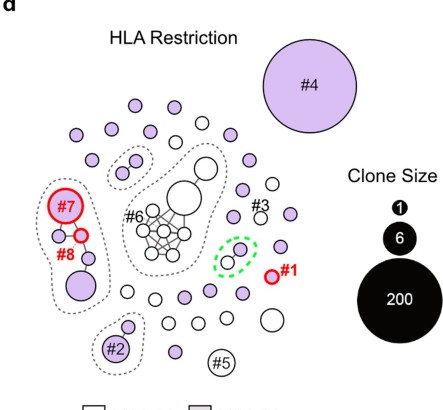

**Extended Data Fig. 4 | Identification, activation, and HLA restriction, of cross-reactive CD8+ T cells. a**, Gating strategy used to identify CD8⁺ T cells which bound to SNX8 epitope and/or MADS N protein epitope (CD8⁺ T cells positive for PE). Representative MIS-C patient and control showing each CD8⁺ T cell which bound to any tetramer (PE⁺) and the relative binding of that T cell to both the SNX8 epitope (BV421⁺) and the MADS N protein epitope (APC⁺) identifying cross-reactive T cells (PE⁺APC⁺BV421⁺). Schematics in panel **a** were created using BioRender (https://www.biorender.com). **b**, Stripplots and boxplots showing percentage of CD8⁺ T cells which are cross-reactive to both SNX8 and MADS in MIS-C patients (*n* = 3) and controls (*n* = 3). Insufficient numbers to perform robust statistical testing. **c**, Stripplots and boxplots showing percentage of total T cells which activate in response to either vehicle (culture media + 0.2% DMSO) or the SNX8 Epitope (SNX8 Epitope (Materials) + culture media + 0.2% DMSO) in MIS-C patients (*n* = 2) and at-risk controls (*n* = 4)

measured by AIM assay. Insufficient numbers to perform robust statistical testing. Dotted line at 3 standard deviations above mean of SNX8 Epitope stimulated controls. **d**, TCRdist Similarity Network of 48 unique, paired TCRαβ sequences (*n* = 259 sequences) obtained from four patients with MIS-C. CD8⁺ T cells were sorted from PBMCs directly *ex vivo* or after 10-days of peptide expansion and staining with A*02:01 or A*02:06 HLA class I tetramers loaded with MADS [LQLPQGITL] and SNX8 [MQMPQGNPL] peptides. Each node represents a unique TCR clonotype. Edges connect nodes with a TCRdist score <150. Dashed lines surround TCR similarity clusters. Node size corresponds to T cell clone size. Nodes are colored based on HLA restriction. TCRs selected for further testing are numbered TCR #1-8. Convergent node circled green. For all boxplots in the figure, the whiskers extend to 1.5 times the interquartile range (IQR) from the quartiles, the boxes represent the IQR, and centre lines represent the median.

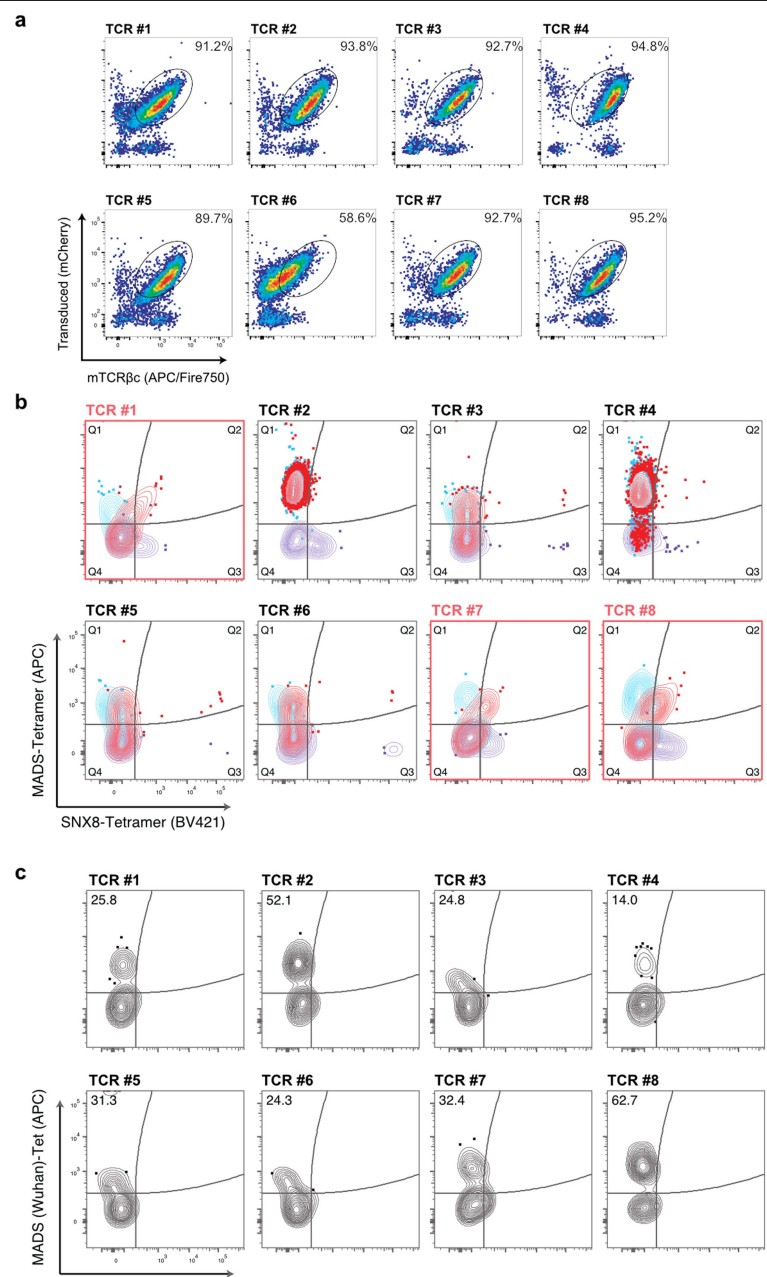

**Extended Data Fig. 5 | Evaluation of Jurkat-TCR lines. a**, Jurkat-76 cells stably expressing putative cross-reactive TCRs (#1-8) stained with anti-murine TCRβ constant region (mTCRβc). Plots depict frequency of transduced (mCherry[+]) Jurkat cells with presence of surface TCR (APC/Fire 750[+]) as a percentage of total live cells. **b**, Jurkat-TCR[+] cell lines expressing putative cross-reactive TCRs #1-8 stained with individual or combination of HLA-A*02:01 or A*02:06 tetramers loaded with MADS [LQLPQGITL] and SNX8 [MQMPQGNPL] peptides. Blue contour plots indicate staining with MADS-Tetramer (PE) and MADS-Tetramer (APC); purple contour plots indicate staining with SNX8-Tetramer (PE) and

SNX8-Tetramer (BV421); red indicates combined staining with a pool of MADS/SNX8-Tetramer (PE), MADS-Tetramer (APC), and SNX8-Tetramer (BV421). Plots shown are gated from total PE[+] cells. Plots with confirmed cross-reactive TCRs outlined in red. **c**, Jurkat-TCR+ cell lines expressing putative cross-reactive TCRs #1-8 stained with individual HLA-A*02:01 or A*02:06 tetramers loaded with MADS Wuhan [LQLPQGTTL] peptide. Gate values indicate frequency of MADS-APC[+] cells as percentage of total MADS-PE[+] cells. Outliers shown in contour plots. Flow plots representative of two independent evaluations.

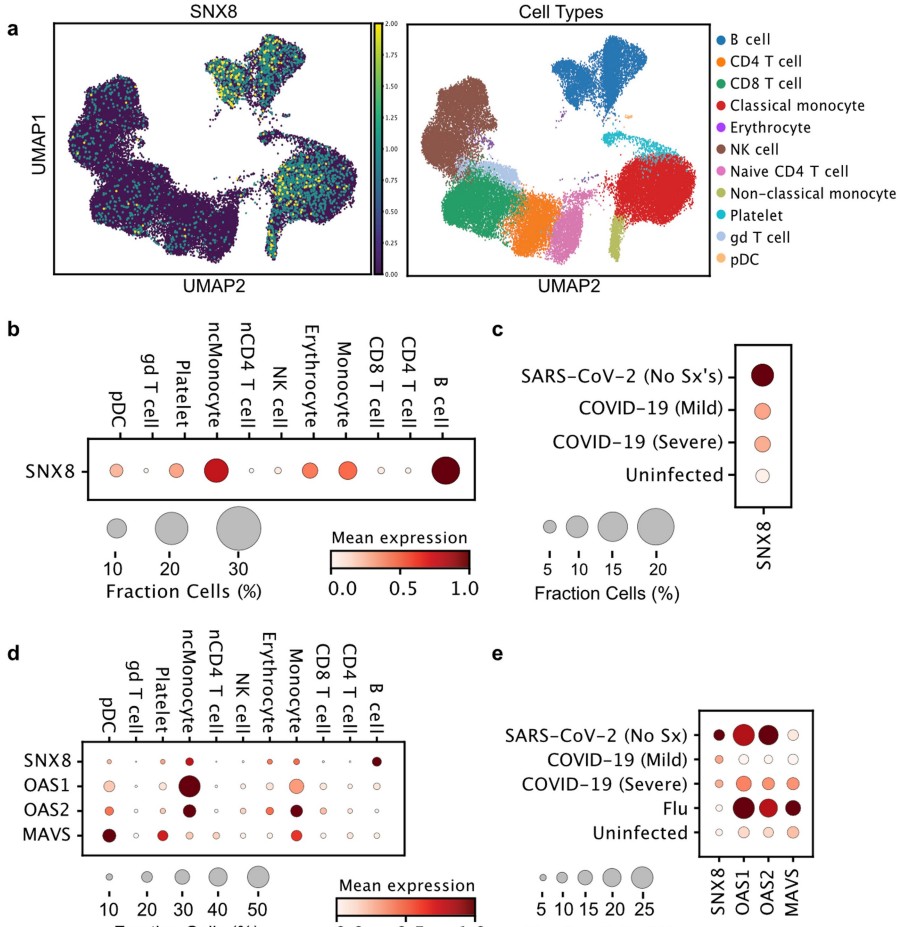

**Extended Data Fig. 6 | *SNX8* expression during viral infection. a**, UMAPs showing *SNX8* expression in various peripheral blood cell types during SARS-CoV-2 infection. **b**, Mean expression and percent of cells expressing *SNX8* in peripheral blood subsets during SARS-CoV-2 infection. **c**, Mean expression and percent of cells expressing *SNX8* averaged across all peripheral blood mononuclear cells from SARS-CoV-2 infected individuals without symptoms, with mild symptoms, or with severe disease compared to uninfected controls. **d**, Mean expression and percent of cells expressing *SNX8*, *OAS1*, *OAS2*, and *MAVS* in peripheral blood subsets during SARS-CoV-2 infection. **e**, Relative expression of *SNX8*, *OAS1*, *OAS2*, and *MAVS* during influenza virus infection compared to different severities of SARS-CoV-2 infection.

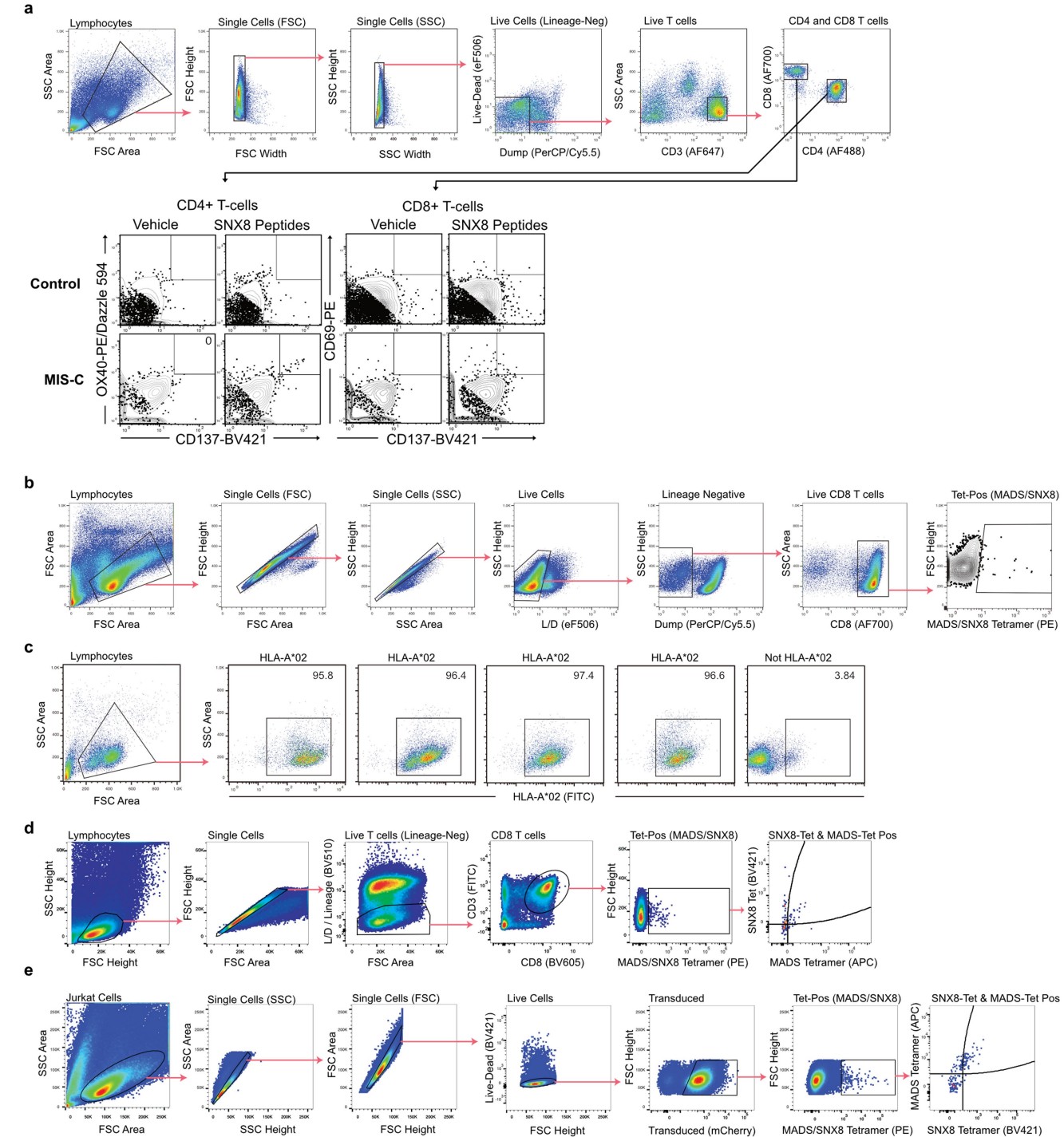

**Extended Data Fig. 7 | Representative flow cytometry gating. a**, Flow cytometry gating strategy for identifying CD4 positive and CD8 positive T cells for the AIM analysis with representative activation induced marker (AIM) assay flow cytometry gating strategy measuring percent of CD4⁺ T cells which activate (CD137⁺OX40⁺) and percent of CD8⁺ T cells which activate (CD137⁺CD69⁺) in response to SNX8 protein. **b**, Flow cytometry gating strategy for the initial SNX8/MADS tetramer cross-reactivity assay (Extended Data Fig. 4a,b) showing isolation of PE-tetramer positive CD8 positive T cells. **c**, Flow cytometry plots showing results of serotyping for the PBMCs used in the initial SNX8/MADS tetramer cross-reactivity assay (Extended Data Fig. 4a,b) which did not have

sufficient cells for genotyping. Shown is the 1 MIS-C patient (far left) and 3 controls (middle 3) which are positive for HLA-A*02 and were used and one control negative for HLA-A*02 (far right) which was not used. **d**, Index sorting strategy for patient PBMCs from *ex vivo* and peptide expansion experiments for TCR sequencing. Single cells were sorted from live/lineage (CD4, CD14, CD16, CD19)-negative, CD3⁺CD8⁺ T lymphocytes positive for MADS/SNX8-Tetramer (PE) and MADS-Tetramer (APC) and/or SNX8-Tetramer (BV421). **e**, Flow cytometry gating strategy to evaluate putative cross-reactive Jurkat-TCRs. Gates include single, live, transduced Jurkat lymphocytes triple positive for MADS/SNX8-(PE), MADS-(APC), and SNX8-(BV421) tetramers shown in Fig. 4.

**Extended Data Table 1 | Clinical characteristics of MIS-C and at-risk control cohorts**

|  | MIS-C | Control |
|---|---|---|
| Number | 199 | 45 |
| Male sex (%) | 118 (59.3) | 24 (53.3) |
| Median age in years (median, IQR) | 10.9 (7.5 – 14.7) | 5.5 (2.2 – 11.1) |
| **Race and ethnicity (%)** |  |  |
| White, non-Hispanic | 63 (31.7) | 14 (31.1) |
| Black, non-Hispanic | 74 (37.2) | 5 (11.1) |
| Hispanic or Latino | 47 (23.6) | 15 (33.3) |
| Other race, non-Hispanic | 8 (4.0) | 2 (4.4) |
| Unknown | 7 (3.5) | 9 (20.0) |
| **Underlying conditions (%)** |  |  |
| None | 134 (67.3) | 36 (80) |
| Obesity* | 29 (15.2) | 8 (21.6) |
| Asthma | 22 (11.1) | 1 (2.2) |
| Cardiovascular | 4 (2.0) | 0 (0.0) |
| Immunocompromised | 0 (0.0) | 0 (0.0) |
| Autoimmune condition | 1 (0.5) † | 0 (0.0) |
| Malignancy | 1 (0.5) | 0 (0.0) |
| **Hospital Course (%)** |  |  |
| ICU admission | 170 (85.4) | N/A |
| Shock requiring vasopressors | 90 (45.2) | N/A |
| Mechanical ventilation | 38 (19.1) | N/A |
| Extracorporeal membrane oxygenation (ECMO) | 8 (4.0) | N/A |

*Does not include *n*=8 MIS-C patients and *n*=11 controls under 2 years of age.

†Patient on chronic systemic steroids for eosinophilic esophagitis.

N/A; not applicable as hospitalization excluded a patient from being part of our control group.

**Extended Data Table 2 | Phenotypic data for MIS-C patients**

| Organ System Involvement (number reporting) | Phenotype Present | Percent |
|---|---|---|
| **Respiratory** | | |
| Oxygen requirement (198) | 139 | 70.2 |
| Bronchospasm (161) | 2 | 1.2 |
| Acute Respiratory Distress Syndrome (ARDS) (198) | 24 | 12.1 |
| **Cardiac** | | |
| Vasoactive support (198) | 124 | 62.6 |
| Myocarditis (199) | 115 | 57.8 |
| Elevated Troponin (197) | 128 | 65 |
| **Neurologic** | | |
| Any neurologic symptoms (197) | 22 | 11.2 |
| Encephalitis (197) | 2 | 1 |
| **Gastrointestinal** (196) | 66 | 33.7 |
| **Hematologic** (197) | 22 | 11.2 |
| **Rheumatologic** | | |
| Arthritis (195) | 9 | 4.6 |
| Myositis (194) | 27 | 12.9 |
| **Mucocutaneous** | | |
| Conjunctivitis (197) | 105 | 53.3 |
| Oral Mucosal Changes (198) | 76 | 38.4 |
| Rash (197) | 106 | 53.8 |

Oxygen requirement; receipt of any oxygen support at any time during hospitalization. Bronchospasm; severe bronchospasm requiring continuous bronchodilators. Acute Respiratory Distress Syndrome (ARDS); onset of hypoxemia was acute (during this illness), chest imaging findings of new infiltrates (unilateral or bilateral), respiratory failure not fully explained by cardiac failure or fluid overload, PaO2/FiO2 ratio < 300 or SpO2/FiO2 < 264 (if SpO2 < 97), on CPAP > 5 cm H2O or BiPAP or Invasive Mechanical Ventilation. Vasoactive support; receipt of vasoactive infusions (at any time during hospitalization) including: Dopamine, Dobutamine, Epinephrine, Norepinephrine, Phenylephrine, Milrinone, Vasopressin (for hypotension, not diabetes insipidus). Myocarditis; myocarditis diagnosed during hospital stay and adjudicated by outside panel of cardiologists. Elevated troponin; based on site-specific cutoff. Any neurologic symptoms; suspected central nervous system infection, stroke or intracranial hemorrhage (at presentation or during hospitalization), seizure (at presentation or during hospitalization), coma or unresponsive, receipt of neurodiagnostic imaging (CT, MRI, or LP), encephalitis, decreased hearing, decreased vision, iritis or uveitis. Gastrointestinal; appendicitis, diarrhea (at presentation or during hospitalization), abdominal pain (at presentation or during hospitalization), gallbladder hydrops or edema, pancreatitis, hepatitis, nausea/loss of appetite at presentation, vomiting at presentation. Hematologic; anemia with hemoglobin <9 g/dL, minimum white blood cells <4 x 10³ cells/μL, minimum platelets <150 x 10³ cells/μL, deep vein thrombosis, pulmonary embolism, hemolysis, bleeding, ischemia of an extremity. Oral Mucosal Changes; erythema of lips or oropharynx, strawberry tongue, or drying or fissuring of the lips.

**Extended Data Table 3 | Clinical characteristics of validation cohorts**

|  | Severe Acute COVID-19 | MIS-C Validation |
|---|---|---|
| Number | 29 | 24 |
| Male sex (%) | 14 (48.3) | 16 (66.7) |
| Median age in years (median, IQR) | 11.4 (11.4) | 7.7 (8.6) |
| **Race and ethnicity (%)** | | |
| White, non-Hispanic | 14 (48.3) | 3 (12.5) |
| Black, non-Hispanic | 4 (13.8) | 8 (33.3) |
| Hispanic or Latino | 9 (31.0) | 13 (54.2) |
| Other race, non-Hispanic | 2 (6.9) | 0 (0.0) |
| **Underlying conditions (%)** | | |
| None | 9 (31.0) | 13 (54.2%) |
| Obesity* | 8 (27.6) | 7 (29.2) |
| Asthma | 3 (10.3) | 2 (8.3) |
| Cardiovascular | 1 (3.4) | 0 (0.0) |
| Immunocompromise | 1 (3.4) | 0 (0.0) |
| Autoimmune condition | 0 (0.0) | 0 (0.0) |
| Malignancy | 1 (3.4) | 0 (0.0) |
| **Interventions (%)** | | |
| ICU admission | 22 (75.9) | 11 (45.8) |
| Shock requiring vasopressors | 11 (37.9) | 0 (0.0) |
| Mechanical ventilation | 6 (20.7) | 2 (8.3) |

*Does not include acute COVID-19 ($n$=4) and MIS-C ($n$=1) patients under 2 years of age.

# Reporting Summary

## Statistics

For all statistical analyses, confirm that the following items are present in the figure legend, table legend, main text, or Methods section.

| n/a | Confirmed | |
|---|---|---|
| ☐ | ☒ | The exact sample size (*n*) for each experimental group/condition, given as a discrete number and unit of measurement |
| ☐ | ☒ | A statement on whether measurements were taken from distinct samples or whether the same sample was measured repeatedly |
| ☐ | ☒ | The statistical test(s) used AND whether they are one- or two-sided *Only common tests should be described solely by name; describe more complex techniques in the Methods section.* |
| ☒ | ☐ | A description of all covariates tested |
| ☐ | ☒ | A description of any assumptions or corrections, such as tests of normality and adjustment for multiple comparisons |
| ☐ | ☒ | A full description of the statistical parameters including central tendency (e.g. means) or other basic estimates (e.g. regression coefficient) AND variation (e.g. standard deviation) or associated estimates of uncertainty (e.g. confidence intervals) |
| ☐ | ☒ | For null hypothesis testing, the test statistic (e.g. *F*, *t*, *r*) with confidence intervals, effect sizes, degrees of freedom and *P* value noted *Give P values as exact values whenever suitable.* |
| ☒ | ☐ | For Bayesian analysis, information on the choice of priors and Markov chain Monte Carlo settings |
| ☒ | ☐ | For hierarchical and complex designs, identification of the appropriate level for tests and full reporting of outcomes |
| ☒ | ☐ | Estimates of effect sizes (e.g. Cohen's *d*, Pearson's *r*), indicating how they were calculated |

*Our web collection on statistics for biologists contains articles on many of the points above.*

## Software and code

Policy information about availability of computer code

| Data collection | RAPSearch2.0 was used to align all Illumina generated PhIP-Seq Fastq files. Flow cytometry data were collected using FACSDiva v8.01 Software (Becton Dickinson) or SpectroFlow v2.2 software (Cytek). |
|---|---|
| Data analysis | Python 3 and R v3.6.0 were used for data analysis. For the machine learning logistic regression classifier, the Scikit-learn Python package was utilized, and is referenced in the "PhIP-Seq Analysis" section of the methods, and a previous publication using this analysis is cited. For TCR sequencing and repertoire analysis, the TCRdist algorithm implementation from the CoNGA v0.1.2 python package was used. Stitchr vl.0.0 was used to reconstruct TCR sequences. Further analysis was performed using R, with merging and subsetting of data performed using the dplyr packages. TCR similarity networks were built using stringdist v0.9.12 and igraph v2.0.3 R packages and visualized using gephi v0.9.7 software. Visualizations in R was performed using ggplot2 v3.4.0 and ggpubr v0.5.0. Cell population gating and fluorescence analysis was performed using FlowJo version 10.7.2 software (BD Biosciences). Any figures created with BioRender.com were exported under a paid subscription with an associated publication license. |

For manuscripts utilizing custom algorithms or software that are central to the research but not yet described in published literature, software must be made available to editors and reviewers. We strongly encourage code deposition in a community repository (e.g. GitHub). See the Nature Portfolio guidelines for submitting code & software for further information.

## Data

Policy information about availability of data

All manuscripts must include a data availability statement. This statement should provide the following information, where applicable:
- Accession codes, unique identifiers, or web links for publicly available datasets
- A description of any restrictions on data availability
- For clinical datasets or third party data, please ensure that the statement adheres to our policy

The published article includes all datasets generated or analyzed as a part of this study. Individual source data are provided with associated figures (where appropriate) per the data sharing agreement stipulated under the Ruth L. Kirschstein National Research Service Award Individual Postdoctoral Fellowship (award no. F32AI157296; R.C.M.). Raw flow cytometry source files can be made available upon reasonable request. Source data are provided with this paper. All PhIP-Seq data are publicly available via a Dryad digital repository: DOI: 10.7272/Q6SJ1HVH. Raw TCR sequencing reads available through NCBI Short Read Archive (SRA) BioProject #PRJNA1110271 with associated BioSample Accession numbers SAMN41334731, SAMN41334732, SAMN41334730, SAMN41334729.

## Research involving human participants, their data, or biological material

Policy information about studies with human participants or human data. See also policy information about sex, gender (identity/presentation), and sexual orientation and race, ethnicity and racism.

| | |
|---|---|
| Reporting on sex and gender | All sex data refers to sex and not gender and is presented in Extended Data Table 1 and Extended Data Table 3. |
| Reporting on race, ethnicity, or other socially relevant groupings | Race and ethnicity were self-reported and presented in Extended Data Table 1 and Extended Data Table 3. |
| Population characteristics | Additional characteristics including age and underlying medical conditions are provided in Extended Data Table 1. Detailed clinical data describing the population during the course of illness is presented in Extended Data Table 2. |
| Recruitment | Patients were recruited through the prospectively enrolling multicenter Overcoming COVID-19 and Taking on COVID-19 Together study in the United States. A total of 292 patients were enrolled into 1 of the following independent cohorts between June 1, 2020 and September 9, 2021: 223 patients hospitalized with MIS-C (199 in the primary discovery cohort, 24 in a separate subsequent validation cohort), 29 patients hospitalized for COVID-19 in either an intensive care or step-down unit (referred to as severe acute COVID-19 in this study), and 45 outpatients (referred to as "at-risk controls" in this study) post-SARS-CoV-2 infections associated with mild or no symptoms. For use as controls in the SARS-CoV-2 specific PhIP-Seq, plasma from 48 healthy, pre-COVID-19 controls were obtained as deidentified samples from the New York Blood Center. These samples were part of retention tubes collected at the time of blood donations from volunteer donors who provided informed consent for their samples to be used for research. We are not aware of any self-selection bias which would alter the results of this study as any patient meeting eligibility criteria at any site was eligible. |
| Ethics oversight | The study was approved by the central Boston Children's Hospital Institutional Review Board (IRB) and reviewed by IRBs of participating sites with CDC IRB reliance. |

Note that full information on the approval of the study protocol must also be provided in the manuscript.

# Field-specific reporting

Please select the one below that is the best fit for your research. If you are not sure, read the appropriate sections before making your selection.

☒ Life sciences        ☐ Behavioural & social sciences        ☐ Ecological, evolutionary & environmental sciences

For a reference copy of the document with all sections, see nature.com/documents/nr-reporting-summary-flat.pdf

# Life sciences study design

All studies must disclose on these points even when the disclosure is negative.

| | |
|---|---|
| Sample size | Sample size was determined based on sample availability, with the goal of including as many samples as possible. This ultimately led to 199 MIS-C patient samples and 45 at-risk control patient samples. Each sample was included in the initial human proteome-wide PhIP-Seq screen. As many samples as possible were included in subsequent experiments, with some attrition as samples from certain individuals were exhausted. The total number of samples used in each experiment is mentioned in the text and figures. For experiments utilizing patient PBMCs, as many samples were utilized as available in our cohort. This ultimately led to experiments conducted on PBMCs from 11 patients with MIS-C and 10 at-risk controls. Given limited number of PBMCs, the activation induced marker assay for identifying SNX8 autoreactive T-cells was prioritized and performed on all patients. Three patients and three controls had a sufficient quantity of PBMCs and correct HLA-type to perform initial tetramer assays, and three additional MIS-C patients were identified in our biobank for use in isolation of the cross-reactive T cell receptors. |
| Data exclusions | The activation induced marker assay for detecting SNX8 autoreactive T-cells was run on 11 patients with MIS-C and 10 controls. Data is only provided for 9 of the 11 MIS-C patients, because 2 of the MIS-C samples had insufficient total flow cytometry events to analyze (total of 5,099 |

and 4,919 events), and is discussed in the methods.

| | |
|---|---|
| Replication | Because all experiments were performed on human samples with limited supplies, we were not able to repeat the same experiment in the same individual except for the essential experiment of confirming SNX8 autoreactivity in patients and controls to the peptide containing the identified epitope. We included as many samples as possible in each experiment such that the cases and controls served as "biologically similar" samples to one another. We also performed extensive orthogonal validation experiments to reproduce key findings with additional assays. Given the limited number of patient PBMCs, repeating AIM and tetramer binding assays, and T cell receptor isolation experiments, was not possible. |
| Randomization | Samples were allocated based on clinical disease category. |
| Blinding | PhIP-Seq was performed with the experimentalists blinded to the samples. Targeted orthogonal validation experiments were not performed blinded to samples, though they were conducted in relatively high throughput with the majority of experiments utilizing 96-well plates with disease categories intermixed making it unlikely the experimenter could be aware of which sample corresponded to which disease category. PhIP-Seq data was analyzed using unbiased, unsupervised methods, but disease category for each sample was known. Targeted immunoprecipitation experiments were analyzed identically in all samples regardless of category. For experiments with patient PBMCs, the experimenter was not blinded to disease state but analysis was performed blinded to patient disease category. |

# Reporting for specific materials, systems and methods

We require information from authors about some types of materials, experimental systems and methods used in many studies. Here, indicate whether each material, system or method listed is relevant to your study. If you are not sure if a list item applies to your research, read the appropriate section before selecting a response.

<table>
<tr><td colspan="2">Materials & experimental systems</td><td colspan="2">Methods</td></tr>
<tr><td>n/a</td><td>Involved in the study</td><td>n/a</td><td>Involved in the study</td></tr>
<tr><td>☐</td><td>☒ Antibodies</td><td>☒</td><td>☐ ChIP-seq</td></tr>
<tr><td>☒</td><td>☐ Eukaryotic cell lines</td><td>☐</td><td>☒ Flow cytometry</td></tr>
<tr><td>☒</td><td>☐ Palaeontology and archaeology</td><td>☒</td><td>☐ MRI-based neuroimaging</td></tr>
<tr><td>☒</td><td>☐ Animals and other organisms</td><td></td><td></td></tr>
<tr><td>☒</td><td>☐ Clinical data</td><td></td><td></td></tr>
<tr><td>☒</td><td>☐ Dual use research of concern</td><td></td><td></td></tr>
<tr><td>☒</td><td>☐ Plants</td><td></td><td></td></tr>
</table>

## Antibodies

| | |
|---|---|
| Antibodies used | Anti-HiBit positive control antibody (Promega, Madison, WI; #CS2006A01 1:10 dilutions), anti-Myc positive control antibody (Cell Signaling Technology, #2272 S 1:10 dilution) were used. FITC anti-human HLA-A2 Antibody (BioLegend #343303, Clone BB7.2; 1:100 dilution), Alexa 647 conjugated anti-CD3 (BioLegend #317312, Clone OKT3; 1:100 dilution), Alexa 488 conjugated anti-CD4 (BioLegend #317420, Clone OKT4; 1:100 dilution), Alexa 700 conjugated anti-CD8 (BioLegend #344724, Clone SK1; 1:100 dilution), PE-Dazzle 594 conjugated anti-OX-40 (BioLegend #350020, Clone ACT35; 1:100 dilution), PE conjugated anti-CD69 (BioLegend # 310906, Clone FN-50; 1:100 dilution), BV421 conjugated anti-CD137 (BioLegend #309820, 4B4-1; 1:100 dilution), PerCP-Cy5.5 conjugated anti- CD14 (BioLegend #325622, Clone HCD14; 1:100 dilution), PerCP-Cy5.5 conjugated anti- CD16 (BioLegend #360712, Clone B73.1; 1:100 dilution), PerCP-Cy5.5 conjugated anti- CD19 (BioLegend #302230, Clone HIB19; 1:100 dilution), eFluor 506 conjugated Live/dead dye (Invitrogen #65-0866-14, 1:100 dilution), PerCP-Cy5.5 conjugated anti-CD4 (BioLegend #300530, Clone RPA-T4; 1:100 dilution). FITC-conjugated anti-human CD3 (Biolegend #317306, clone OKT3, lot# B390808; 1:20 dilution), BV605-conjugated anti-human CD8 (Biolegend #344742, clone SK1, lot# B371925; 1:20 dilution), BV510-conjugated anti-human CD4 (Biolegend #317444, clone OKT4, lot# B375526; 1:20 dilution), BV510-conjugated anti-human CD14 (Biolegend #367124, clone 63D3, lot# B390770; 1:20 dilution), BV510-conjugated anti-human CD16 (Biolegend #302048, clone 3G8, lot# B372132; 1:20 dilution), BV510-conjugated anti-human CD19 (Biolegend #302242, clone HIB19, lot# B390665; 1:20 dilution), and Ghost Dye Violet 510 Viability Dye (Tonbo Biosciences #13-0870-T500, lot# D0870061322133; 1:400 dilution). PE-conjugated anti-human β2m antibody (Santa Cruz Biotech #sc-13565, clone BBM.1) at 1:200. |
| Validation | All antibodies were purchased from commercial suppliers including Promega, Cell Signaling Technology, BD, BioLegend, Tonbo, ThermoFisher, Sigma, and eBiosciences with validation data and applicable citations available on product listings for all antibodies (see individual catalog numbers). Antibodies that have previously been validated in the literature were preferred and used at specified dilutions or according to the manufacturer's specifications. |

# Flow Cytometry

## Plots

Confirm that:

☒ The axis labels state the marker and fluorochrome used (e.g. CD4-FITC).

☒ The axis scales are clearly visible. Include numbers along axes only for bottom left plot of group (a 'group' is an analysis of identical markers).

☒ All plots are contour plots with outliers or pseudocolor plots.

☒ A numerical value for number of cells or percentage (with statistics) is provided.

## Methodology

**Sample preparation**

For AIM assay:
Peripheral blood mononuclear cells (PBMCs) were obtained from 10 patients with MIS-C and 10 controls for use in the AIM assay. PBMCs were thawed, washed, resuspended in serum-free RPMI medium, and plated at a concentration of 1e106 cell/well in a 96-well round-bottom plate. For each individual, PBMCs were stimulated for 24-hours with either the SNX8 pool (see above) at a final concentration of 1 ug/mL/peptide in 0.2% DMSO, or a vehicle control containing 0.2% DMSO only. For 4 of the controls and 2 of the MIS-C patients, there were sufficient PBMCs for an additional stimulation condition using the SNX8 high resolution epitope pool (see above) also at a concentration of 1 ug/mL/peptide in 0.2% DMSO for 24-hours. Following the stimulation, cells were washed with FACS buffer (Dulbecco's PBS without calcium or magnesium, 0.1% sodium azide, 2 mM EDTA, 1% FBS) and stained with the following antibody panel for 20 minutes at 4 degrees and then flow cytometry analysis was immediately performed.

For tetramer assay:
PBMCs from 2 MIS-C patients with HLA-A*02:01 (both PAXGene genotyped, 1 confirmed by serotyping) and 1 MIS-C patient with HLA-B*35.01 (PAXGene genotyped), and 3 at-risk controls with HLA-A*02:01 (all 3 identified by serotyping, 2 of 3 confirmed by PAXGene genotyping, other sample did not have gDNA available for genotyping) were thawed, washed, and put into culture with media containing recombinant human IL-2 at 10 ng/mL in 96-well plates. Peptide fragments LQLPQGITL and MQMPQGNPL were then added to PBMCs to a final concentration of 10 ug/mL/peptide and incubated (37C, 5% CO2) for 7 days.

Following the 7 days of incubation, a total of 8 pMHCI tetramers were generated from UV-photolabile biotinylated monomers, 4 each from HLA-A*02:01 and HLA-B*35:01 (NIH Tetramer Core). Peptides were loaded via UV peptide exchange. Tetramerization was carried out using streptavidin conjugated to fluorophores PE and APC or BV421 followed by quenching with 500uM D-biotin. Tetramers were then pooled together as shown below. All PBMCs were then treated with 100 nM Dasatinib (StemCell) for 30 min at 37 °C followed by staining (no wash step) with the respective tetramer pool corresponding to their HLA restriction (final concentration, 2 to 3 µg/ml) for 30 min at room temperature. Cells were then stained with the cell surface markers for 20 minutes, followed by immediate analysis on a flow cytometer.

For single-cell index sorting of ex vivo and peptide-specific expansion: Unexpanded PBMCs (direct ex vivo) or peptide-expanded T cells were washed in 1x PBS, and treated with 100 nM dasatinib (Sigma-Aldrich #CDS023389) in 1x PBS for 30 min at 37°C and 5% CO2. Cells were then pelleted and resuspended in 50 µL FACS buffer (1x PBS, 0.04% BSA) supplemented with human TruStain FcX blocking buffer (Biolegend #422302; 1:10 dilution), 500 µM D-biotin (ThermoFisher Scientific #B20656), and a unique tetramer cocktail containing MADS-Tetramer-PE (1:10 dilution), MADS-Tetramer-APC (1:10 dilution), SNX8-Tetramer-PE (1:10 dilution) and SNX8-Tetramer-BV421 (1:10 dilution) based on participant HLA type (A*02:01; A*02:06). Cells were incubated in the dark at 25°C for 1 h followed by direct addition of 50 µL (100 µL total volume) of FACS supplemented with 500 µM D-biotin and an antibody cocktail containing FITC-conjugated anti-human CD3 (Biolegend #317306, clone OKT3, lot# B390808; 1:20 dilution), BV605-conjugated anti-human CD8 (Biolegend #344742, clone SK1, lot# B371925; 1:20 dilution), BV510-conjugated anti-human CD4 (Biolegend #317444, clone OKT4, lot# B375526; 1:20 dilution), BV510-conjugated anti-human CD14 (Biolegend #367124, clone 63D3, lot# B390770; 1:20 dilution), BV510-conjugated anti-human CD16 (Biolegend #302048, clone 3G8, lot# B372132; 1:20 dilution), BV510-conjugated anti-human CD19 (Biolegend #302242, clone HIB19, lot# B390665; 1:20 dilution), and Ghost Dye Violet 510 Viability Dye (Tonbo Biosciences #13-0870-T500, lot# D0870061322133; 1:400 dilution) for 30 minutes in the dark at 4°C. Cells were then pelleted, washed twice with 4 mL FACS buffer (containing 500 µM D-biotin), suspended in 500 µL FACS (containing 500 µM D-biotin), and passed through a 45 µM filter before proceeding to single-cell sorting on a Sony SY3200 cell sorter.

For specificity validation of Jurkat-TCR cell lines: 1x10^6 Jurkat-TCR+ cell lines or untransduced Jurkat J76.7 (TCR-null) were washed in 1x PBS and resuspended in 50 µL FACS buffer (1x PBS, 0.04% BSA) and a unique tetramer cocktail containing MADS-Tetramer-PE (1:10 dilution), MADS-Tetramer-APC (1:10 dilution), SNX8-Tetramer-PE (1:10 dilution), and SNX8-Tetramer-BV421 (1:10 dilution) based on the restricting HLA type (A*02:01; A*02:06). Tetramers conjugated to the Wuhan peptide sequence (LQLPQGTTL), including Wuhan-Tetramer-PE (1:10 dilution) and Wuhan-Tetramer-APC (1:10 dilution), were also tested. A second set of wells were set up in which each individual tetramer was used to stain cells. Cells were incubated in the dark at 25°C for 30 min after which 50 uL of FACS buffer containing Ghost Dye Violet 510 Viability Dye (Tonbo Biosciences #13-0870-T500, lot# D0870061322133; 1:400 dilution) was added for an additional 30 min incubation in the dark at 25°C. Cells were then washed twice with 1 mL FACS buffer and suspended in 300 µL FACS.

For HLA monomer fold testing: Unfolded, biotinylated easYmer monomers (Immudex) were obtained for HLA-A*02:01 and HLA-A*02:06. SARS-CoV-2 MADS (LQLPQGITL), SARS-CoV-2 Wuhan (LQLPQGTTL), and human SNX8 (MQMPQGNPL) peptides were commercially synthesized (Genscript), diluted to 1 mM in ddH2O or DMSO, and loaded onto each easYmer allele according to the manufacturer's instructions at 18°C for 48 h. Proper peptide-HLA monomer formation and MADS and SNX8 peptide binding strength was evaluated for each HLA using a 'β2m fold test' relative to negative (no peptide; unloaded

monomer) and positive (strong binding peptide; CMV pp65 495-503 [NLVPMVATV]) controls as per the manufacturer's protocol. Briefly, peptide-loaded monomers with a concentration of 500 nM were serially diluted to 9 nM, 3 nM, and 1 nM in dilution buffer (1x PBS with 5% glycerol [Sigma-Aldrich #G5516]) and incubated with streptavidin beads (Spherotech #SVP-60-5, 6-8 µm) at 37°C for 1 hour to allow binding of stable complexes to beads, then washed three times with FACS buffer (1x PBS, 0.5% BSA [Sigma-Aldrich #A7030], 2 mM EDTA [ThermoFisher Scientific #15575-038]). Samples were then stained with PE-conjugated anti-human β2m antibody (Santa Cruz Biotech #sc-13565, clone BBM.1) at 1:200 for 30 min at 4°C, washed 3 times with FACS buffer, and analyzed on a 5 Laser 16UV-16V-14B-10YG-8R AURORA spectral cytometer (Cytek).

| | |
|---|---|
| Instrument | BD LSR Fortessa; Sony SY3200 cell sorter; 5 Laser 16UV-16V-14B-10YG-8R AURORA spectral cytometer (Cytek) |
| Software | FlowJo v10.7.2 or v10.8.2 was used for the analysis. |
| Cell population abundance | The final cell population abundance is outlined in Figure 4 and Extended Data Figure 6. Cell population frequency (%parent gate) is detailed in Figure 5b and Extended Data Fig. 10a-c |

Gating strategy

For AIM assay:
An initial generous gate was drawn which captured lymphocytes using FSC-A/SSC-A as shown in Extended Data Figure 1A. Singlets were then identified with a FSC-H/FSC-W gate followed by a SSC-H/SSC-W gate. Live cells which were negative for the CD14/CD16/CD19 dump were then gated. T cells were then identified by CD3 surface staining with a clear discrete CD3+ population. CD4 and CD8 cells were then gated on with clear discrete populations. Activated CD4 T-cells were defined as those which were co-positive for OX40 and CD137. Activated CD8 T-cells were defined as those which were co-positive for CD69 and CD137. Gating thresholds for activation were defined by the outer limits of signal in the vehicle controls allowing for up to 2 outlier cells.

For tetramer assay:
An initial generous gate was drawn which captured lymphocytes using FSC-A/SSC-A as shown in Extended Data Figure 1B. Singlets were then identified with a FSC-H/FSC-A gate followed by a SSC-H/SSC-A gate. Dead cells were excluding use a live/dead stain, and CD14/CD16/CD19 positive cells were excluded using a dump gate. CD8 positive surface staining then identified a clear distinct positive population on which to perform the tetramer gating. A stringent tetramer gating strategy was used to identify cross-reactive T-cells, whereby CD8+ T-cells were required to be triple-positive for PE, APC, and BV421 labels (i.e. a single CD8 T-cell bound to PE conjugated LQLPQGITL and/or PE conjugated MQMPQGNPL in addition to APC-conjugated LQLPQGITL and BV421 conjugated MQMPQGNPL). To accomplish this first all PE positive cells were gated on based on identification of outliers from the main CD8+ population. Then a co-positive BV421/APC were identified with an arbitrary gate (insufficient PE+ cells to draw a gate based on distinct cell populations) which was consistent across all samples.

For single-cell index sorting:  Single cells were sorted from live (BV510-neg) and lineage (CD4-BV510, CD14-BV510, CD16-BV510, CD19-BV510)-negative, CD3-FITC+/CD8-BV605+ T lymphocytes positive for MADS/SNX8-Tetramer (PE) and MADS-Tetramer (APC) and/or SNX8-Tetramer (BV421) as described in Extended Data Fig. 9a.

For evaluating Jurkat-TCRs: Gates include single, live (BV510-neg), transduced (mCherry+) Jurkat lymphocytes triple positive for MADS/SNX8-(PE), MADS-(APC), and SNX8-(BV421) tetramers as described in Extended Data Fig. 9b.

☒ Tick this box to confirm that a figure exemplifying the gating strategy is provided in the Supplementary Information.

