## [Peer Review file · Nature]

Manuscript Title: Molecular mimicry in multisystem inflammatory syndrome in children (MIS-C)

Reviewer Comments & Author Rebuttals

Reviewer Reports on the Initial Version:

Referees' comments:

Referee #1 (Remarks to the Author):

Bodansky and colleagues use phip-seq as well as orthogonal validation experiments to identify potential antigens that may be implicated in MIS-C.

This is important and well-written work which provides new leads towards a mechanistic understanding of MIS-C.

See my mostly clarifying comments below:

- you have 199 case and 45 control samples. Can you comment on statistical power given that you profile a very large number of antigens?
- this is an unbalanced (199 vs 45) dataset in terms classification. therefore, the random accuracy is not 50%. This should be mentioned.
- 1c: given that performed the cross-validation many times, why are there no error bars in the 1c?
- line 193: can you comment on why you don't find overlap with previous literature?
- line 256: can you comment on why performance drops on the validation set?

Referee #2 (Remarks to the Author):

The authors investigate the role of antibody and T cell cross-reactivity in the multisystem inflammatory syndrome in children (MIS-C) caused by prior SARS-CoV2 infection. They identified antibodies against several host proteins present at higher levels in MIS-C compared to control cases, including Abs to SNX8, ERFL and KDELR1. Furthermore, they determined an epitope of SNX8 with high sequence similarity to the viral N protein. Finally, they were able to demonstrate T cell cross-reactivity involving SNX8 and the N protein, in particular for HLA-A2 positive patients. These results represent a significant advance in understanding the role of autoreactivity in MIS-C. Strengths of this study include the large cohort of MIS-C patients (n=199) and systematic mapping of Ab responses across the human proteome with PhIP-seq. Furthermore, the authors were able to demonstrate both antibody and T cell cross-reactivity against the identified regions of SNX8 and the viral N protein. Overall, the technical quality of the data is high, but a number of important issues remain to be addressed.

1. All three identified proteins represent intracellular antigens, and it is thus unlikely that the Abs have a direct role in disease pathogenesis. Cross-reactive T cells could contribute to the inflammatory syndrome, but in most patients the frequency of such cross-reactive T cells was low. It

remains unclear why T cells present at such a low frequency would cause such a severe inflammatory syndrome. Also, SNX8 antibodies are only present in a subset of cases (17% with the stringent cutoff chosen by the authors).

2. The authors used a rigorous tetramer labeling approach to identify CD8 T cells that cross-reacted with peptides from SNX8 and N protein. However, the number of analyzed MIS-C cases (n=3) was small. Furthermore, the frequency of detected cross-reactive T cells was low (between 0 – 0.15%, Fig. 4D) even though the cells were stimulated in vitro for 1 week with the identified peptides. Such in vitro stimulation should result in very substantial expansion of such T cells. Does this mean that the frequency of cross-reactive CD8 T cells is very low? If that is the case, what are the implications for a pathogenic role of such cross-reactive T cells in this severe inflammatory disease?

3. While the authors demonstrate the presence of cross-reactive T cells, their functional programs remain unknown. It would be highly instructive to provide evidence that such cross-reactive T cells are in an activated, inflammatory state in vivo. An approach could be to combine scRNA-seq (including TCR sequencing) with the tetramer labeling approach shown in Figure 4.

4. The affinity of the peptides from SNX8 and viral N protein needs to be experimentally determined. The authors only show predicted binding to HLA-A2. This question matters because low affinity binding of the SNX8 peptide by HLA-A2 would substantially reduce the density of this HLA-peptide complex on target cells.

5. No data are presented on functional consequences of CD8 T cell recognition of target cells that express the SNX8 epitope. Do such T cells kill monocytes or B cells expressing SNX8? Do they secrete proinflammatory cytokines that could contribute to MIS-C?

Referee #3 (Remarks to the Author):

In this manuscript Bodansky and colleagues set out to examine autoimmune responses in a relatively large cohort of MIS-C patients. The authors used PhiP -seq and identified a 9-mer B cell epitope in SNX8 that likely cross- reacts with a broadly similar 9-mer in the SARS-CoV-2 N protein. Not surprisingly some CD4+ T cells that recognize SNX8 peptides on an AIM assay were also identified. Some CD8+ T cells that are cross-reactive for the 9-mer peptide from SNX8 and the similar 9-mer peptide from the N-protein were identified using MHC Class I-peptide tetramers.

The data are potentially interesting but incomplete. It is theoretically possible that MIS-C in some patients is linked to a transient auto-reactive CD8+ cytotoxic T cell response against the MQMPQGNPL peptide in SNX8. However, there is no data provided to support this view in any causal sense, and if such data existed that would transform this description into a fascinating and important story.

While transient autoimmune phenomena are generally not broadly relevant from an autoimmune disease context after infection, (though they are certainly relevant in the distinct context of checkpoint blockade - wherein they can even be life threatening), they could indeed be causal in contexts like pre-pandemic Kawasaki syndrome and in MIS-C. If there was strong evidence provided for causality in MIS-C, this manuscript would indeed then have been considered very significant.

Finding autoreactivity, especially driven by cross-reactivity, after infection is “normal”. Transient autoimmune phenomena are a part of every infection. A large portion of the B cell repertoire is autoreactive and a large fraction of TFH and “pre-TFH” cells are known to be autoreactive at all times in mice and humans (see PMID: 36759711 for an example). In the context of SARS-CoV-2 infection there has even been a demonstration that BCRs that recognize autoantigens are the same BCRs that recognize SARS-CoV-2 epitopes (PMID: 36044993). Infection can activate cross-reactive pre-existing T cells that were not eliminated by central tolerance; infection not only triggers pre-existing autoreactive T cells but also transiently disrupts homeostatic T-reg mediated regulation of Tcon cells. The link to autoreactive B cells and CD4+ T cells has been seen before. A causally relevant CD8+ T cell phenomenon would be of interest.

Given the possible linkage of MIS-C to certain inherited HLA haplotypes, finding an association of specific auto-antigens, auto-antibodies, autoreactive CD4+ T cells and autoreactive CD8+ T cells in subsets of MIS-C patients compared to patients with COVID-19 alone may not be too surprising. The CD8+ T cell tetramer studies were performed after in vitro expansions of CD8+ T cells with peptide stimulation – but are there naturally occurring clonal expansions of CD8+ T cells in active MIS-C?

The following data would make this study of sufficient interest.

1. Evidence for natural clonal expansion of CD8+ T cells in the blood of MIS-C patients that cross-react with the relevant SNX8 nonamer.
2. Showing that these clonally expanded CD8+ T cells can damage tissues that are known tissue targets in MIS-C

Author Rebuttals to Initial Comments:

Response to Referees' comments:

Referee #1 (Remarks to the Author):

Bodanksy and colleagues use phip-seq as well as orthogonal validation experiments to identify potential antigens that may be implicated in MIS-C.

This is important and well-written work which provides new leads towards a mechanistic understanding of MIS-C.

See my mostly clarifying comments below:

- you have 199 case and 45 control samples. Can you comment on statistical power given that you profile a very large number of antigens?

The sample size of this data set was determined by the total number of patients and controls enrolled in the study. To account for the very large number of antigens profiled, we utilized Bonferroni false discovery rate adjustments with a more stringent alpha cutoff of 0.01 to determine our "Enrichment Set" and "MIS-C Set" (lines 148-152, 190-192).

– this is an unbalanced (199 vs 45) dataset in terms classification. therefore, the random accuracy is not 50%. This should be mentioned.

We agree this is an unbalanced dataset and the random accuracy is therefore not 50%. We have clarified this in the manuscript (lines 186-189).

- 1c: given that performed the cross-validation many times, why are there no error bars in the 1c?

Thank you for pointing this out. We have now updated the figure and legend to include the error bars from 1,000 iterations of the logistic regression (lines 156-159). However, the logistic regression values are very consistent, making the error bars small and potentially difficult to visualize (Fig. 1c).

-line 193: can you comment on why you don't find overlap with previous literature?

In fact, we do find enrichment for nearly all previous literature reported autoantigens in our MIS-C patients, however we also find enrichment for these targets in our at-risk control cohort. Because our control cohort of post-covid pediatric patients is substantially larger than previous studies, we are better able to remove false positives not actually specific for MIS-C. All previously identified autoantigens were thus eliminated, except for UBE3A and the special case of IL1Ra. UBE3A was ultimately not considered because of the presence of signal in 2 convalescent controls. For IL1Ra, MIS-C specific enrichment was not significant after controlling for patients that had received IVIg, a significant confounder for autoimmune detection strategies (lines 215-218). We have added additional clarification (lines 200-205).

- line 256: can you comment on why performance drops on the validation set?

Thank you for this comment. Overfitting in machine learning is a common feature, and thus ROC values for independent validation sets are rarely as robust as the original training set, even with cross validation. This highlights the importance of evaluating independent validation sets. Also, the validation set is significantly smaller (n=24) versus the original training set (n=199). Considering these factors, a drop from 0.93 to 0.84 was not unanticipated.

Referee #2 (Remarks to the Author):

The authors investigate the role of antibody and T cell cross-reactivity in the multisystem inflammatory syndrome in children (MIS-C) caused by prior SARS-CoV2 infection. They identified antibodies against several host proteins present at higher levels in MIS-C compared to control cases, including Abs to SNX8, ERFL and KDELR1. Furthermore, they determined an epitope of SNX8 with high sequence similarity to the viral N protein. Finally, they were able to demonstrate T cell cross-reactivity involving SNX8 and the N protein, in particular for HLA-A2 positive patients. These results represent a significant advance in understanding the role of autoreactivity in MIS-C. Strengths of this study include the large cohort of MIS-C patients (n=199) and systematic mapping of Ab responses across the human proteome with PhIP-seq. Furthermore, the authors were able to demonstrate both antibody and T cell cross-reactivity against the identified regions of SNX8 and the viral N protein. Overall, the technical quality of the data is high, but a number of important issues remain to be addressed.

1. All three identified proteins represent intracellular antigens, and it is thus unlikely that the Abs have a direct role in disease pathogenesis. Cross-reactive T cells could contribute to the inflammatory syndrome, but in most patients the frequency of such cross-reactive T cells was low. It remains unclear why T cells present at such a low frequency would cause such a severe inflammatory syndrome. Also, SNX8 antibodies are only present in a subset of cases (17% with the stringent cutoff chosen by the authors).

Yes, as the Reviewer notes, we do not detect antibody autoreactivity to SNX8 in all MIS-C patients, at least by this assay. Another limitation is that for T cells, we tested binding to a single epitope. This leaves the possibility that other peptides from SNX8 or N protein may be bound by additional T cells, making a frequency determination difficult. Despite these limitations, the newly included *ex vivo* tetramer staining data (in new Figure 5) suggest that the cross-reactive cells are present at a minimum frequency of 1 in 25,000 CD8+ T cells, at least in these patients.

To our knowledge, it remains unclear what frequency of autoreactive CD8+ T cells are required to produce a systemic inflammatory syndrome such as MIS-C, since no such cells have been previously characterized. Although imperfect, we do note that a frequency of 1:25,000 exceeds the number of antigen specific T cells observed in the peripheral blood of patients with other autoimmune diseases¹⁻³. Furthermore, we cannot rule out the potential that the abundance of antigen-specific T cells is greater in the tissue than periphery². We have added discussion text to address these important points (lines 600-615).

2. The authors used a rigorous tetramer labeling approach to identify CD8 T cells that cross-reacted with peptides from SNX8 and N protein. However, the number of analyzed MIS-C cases (n=3) was small. Furthermore, the frequency of detected cross-reactive T cells was low (between 0 – 0.15%, Fig. 4D) even though the cells were stimulated *in vitro* for 1 week with the identified peptides. Such *in vitro* stimulation should result in very substantial expansion of such T cells. Does this mean that the frequency of cross-reactive CD8 T cells is very low? If that is the case, what are the implications for a pathogenic role of such cross-reactive T cells in this severe inflammatory disease?

Similar to our response above, we agree that our previous tetramer labeling approach does not determine the frequency of cross-reactive T-cells *in vivo*. However, we repeated our tetramer staining assay (using minimum epitopes for SNX8 and the MIS-C associated domain of SARS-CoV-2 (MADS)) directly on *ex vivo* samples without expansion. This approach allowed us to detect clonally expanded cross-reactive CD8+ T cells with an estimated minimum frequency of 1 in 25,000. Further, the AIM assay was also performed on patient PBMCs *ex vivo* using a larger set of tiled peptides, and therefore should be more reflective of the number of SNX8 autoreactive T-cells in circulation. We found that approximately 1 in 1,000 peripheral T cells were autoreactive to SNX8, which we believe indeed represents a considerable frequency of autoreactive T-cells. This frequency of autoreactive T cells is at least 10 to 100-fold higher than what is typically seen in peripheral blood of other known autoimmune diseases, including Type 1 Diabetes Mellitus and multiple sclerosis¹⁻³, and is on par with the frequency of viral antigen-specific CD8 T cells seen during acute infections.

3. While the authors demonstrate the presence of cross-reactive T cells, their functional programs remain unknown. It would be highly instructive to provide evidence that such cross-reactive T cells are in an activated, inflammatory state *in vivo*. An approach could be to combine scRNA-seq (including TCR sequencing) with the tetramer labeling approach shown in Figure 4.

We agree that a scRNA-seq approach (including TCR sequencing) of these cross-reactive CD8 T cells would be informative if these cells were of sufficient frequency in peripheral circulation from *ex vivo* patient samples to draw meaningful conclusions. However, because we previously only identified these cross-reactive CD8 T cells following peptide stimulation, we were concerned they may be exceedingly rare in *ex vivo* samples. Furthermore, given that MIS-C is a rare and life-threatening pediatric disease, feasibility of experiments is limited by both the number of patients with available remaining samples and the number of viable cells within each aliquot.

Given the limited sample availability, we opted for a plate-seq approach optimized specifically for TCR sequencing, at the expense of transcriptome-wide sequencing. Through these experiments, we have now identified, cloned, expressed, and characterized cross-reactive TCR sequences with validated specificity for MADS (N protein) and SNX8. Two of the cross-reactive TCRs (#7 and #8 in the figure below) were detected directly *ex vivo* from a patient blood sample and formed a cluster with several other related TCRs based on a calculated TCR similarity metric. Further, two of the TCRs in this same cluster were clonally expanded suggesting recent

recognition and response to SNX8 or SARS-CoV-2 N peptides. These data are summarized in new Figure 5a (see a TCR similarity metric below).

Above is a panel from the **new Figure 5** depicting a paired-chain TCR similarity network calculated by TCRdist from putative cross-reactive TCRs. This TCR network shows clustering of two validated ex vivo-derived cross-reactive TCRs (#7 and #8) with three other ex vivo-derived TCRs. Two clonal expansions (relative node size; $n=6$) were also observed within the cluster suggesting recent expansion/activation. Outlined in green is a cluster of two similar TCRs obtained from ex vivo sampling of different subjects (Patients #2 and #4) with different HLA types which suggests converging TCR responses to similar antigens across individuals.

4. The affinity of the peptides from SNX8 and viral N protein needs to be experimentally determined. The authors only show predicted binding to HLA-A2. This question matters because low affinity binding of the SNX8 peptide by HLA-A2 would substantially reduce the density of this HLA-peptide complex on target cells.

Yes, we agree that determination of the peptide-HLA binding affinity will be an important next step. Here, we leveraged an HLA $\beta 2m$ stability assay as a proxy⁴. As strength of peptide binding is correlated with pHLA- $\beta 2m$ complex stability, increased detection of properly folded $\beta 2m$ has been previously used as a correlate of overall binding affinity. As summarized below and in our new Extended Data Figure 7c, both HLA-A2 alleles (A*02:01 and A*02:06) sufficiently bound both MADS and SNX8 peptides, while HLA-B35 did not bind the MADS and SNX8 peptides. These data are summarized below and in new Extended Data Figure 7c.

5. No data are presented on functional consequences of CD8 T cell recognition of target cells that express the SNX8 epitope. Do such T cells kill monocytes or B cells expressing SNX8? Do they secrete proinflammatory cytokines that could contribute to MIS-C?

In vivo functional consequences of these cross-reactive T cells will be the focus of future studies and remains beyond the scope of this study. In this work, we focused on the primary identification the cross-reactive specificity of the TCRs and their functional validation.

Referee #3 (Remarks to the Author):

In this manuscript Bodansky and colleagues set out to examine autoimmune responses in a relatively large cohort of MIS-C patients. The authors used PhiP -seq and identified a 9-mer B cell epitope in SNX8 that likely cross- reacts with a broadly similar 9-mer in the SARS-CoV-2 N protein. Not surprisingly some CD4+ T cells that recognize SNX8 peptides on an AIM assay were also identified. Some CD8+ T cells that are cross-reactive for the 9-mer peptide from SNX8 and the similar 9-mer peptide from the N-protein were identified using MHC Class I-peptide tetramers.

The data are potentially interesting but incomplete. It is theoretically possible that MIS-C in some patients is linked to a transient auto-reactive CD8+ cytotoxic T cell response against the MQMPQGNPL peptide in SNX8. However, there is no data provided to support this view in any causal sense, and if such data existed that would transform this description into a fascinating and important story.

While transient autoimmune phenomena are generally not broadly relevant from an autoimmune disease context after infection, (though they are certainly relevant in the distinct context of checkpoint blockade - wherein they can even be life threatening), they could indeed be causal in contexts like pre-pandemic Kawasaki syndrome and in MIS-C. If there was strong evidence provided for causality in MIS-C, this manuscript would indeed then have been considered very significant.

Finding autoreactivity, especially driven by cross-reactivity, after infection is “normal”. Transient autoimmune phenomena are a part of every infection. A large portion of the B cell repertoire is

autoreactive and a large fraction of TFH and “pre-TFH” cells are known to be autoreactive at all times in mice and humans (see PMID: 36759711 for an example). In the context of SARS-CoV-2 infection there has even been a demonstration that BCRs that recognize autoantigens are the same BCRs that recognize SARS-CoV-2 epitopes (PMID: 36044993). Infection can activate cross-reactive pre-existing T cells that were not eliminated by central tolerance; infection not only triggers pre-existing autoreactive T cells but also transiently disrupts homeostatic T-reg mediated regulation of Tcon cells. The link to autoreactive B cells and CD4+ T cells has been seen before. A causally relevant CD8+ T cell phenomenon would be of interest.

We acknowledge the reviewer’s insights regarding the development of transient autoimmunity in the setting of infections, including SARS-CoV-2. Our findings significantly diverge from these previous studies of infection associated autoimmune phenomena because rather than showing a broad loss of tolerance, we identify a small set of highly specific (and in the case of SNX8, epitope specific) autoreactivities in MIS-C relative to outpatient SARS-CoV-2 infections without MIS-C. Proving causality using correlation-based analyses alone is difficult, but causality can be inferred by establishing associations that predict a specific pathological outcome from within a population of similarly situated individuals (e.g. recent methods for genetics-based causal inference).

Indeed, this is exactly the kind of approach we have taken here. The autoreactivities we identified do not resolve with acute infection, but instead are present 2-8 weeks following infection and so strongly associate with a specific disease manifestation (MIS-C) that just 3 (novel) autoantigens can effectively classify the disease in contrast to the broad reactivities this reviewer references. Moreover, the effectiveness of this classifier was further validated in an entirely separate cohort which also included children with severe COVID-19, further proving that these autoantigens are indeed MIS-C specific rather than a consequence of general loss of tolerance in the setting of varying severities of SARS-CoV-2 infection.-This type of evidence is, in the human setting and in the absence of a robust animal model or precision intervention, the gold standard for causal inference.

We agree with the reviewer that it has been previously shown that autoreactive BCRs can also recognize SARS-CoV-2 antigens. Notably, we show a novel specificity and narrowness of the response in MIS-C. Our work builds significantly on previous studies (including PMID: 36044993) by showing not just that autoreactive B cells can engage SARS-CoV-2, but that in a post-infectious disease-specific context (MIS-C) B cells preferentially engage an entirely distinct epitope of the SARS-CoV-2 N protein than children who do not develop MIS-C. Again, we believe this strongly implicates the novel responses we report in the etiology of disease. Rather than describing a general phenomenon of cross-reactive BCRs in MIS-C, we identify a single epitope shared amongst numerous individuals with MIS-C that distinguishes patients from children without MIS-C. We also note that the convergence of this N protein epitope in MIS-C with the SNX8 epitope (also highly specific to MIS-C) provides preliminary evidence of molecular mimicry in MIS-C which even if not expanded to include data containing a cross-reactive CD8+ TCR (which we now have) would itself be on par with previous seminal examples of putative molecular mimicry in other diseases such as multiple sclerosis and Guillain-Barre syndrome.

We appreciate the reviewer for highlighting the need to frame our findings in this way and have edited the text accordingly.

Given the possible linkage of MIS-C to certain inherited HLA haplotypes, finding an association of specific auto-antigens, auto-antibodies, autoreactive CD4+ T cells and autoreactive CD8+ T cells in subsets of MIS-C patients compared to patients with COVID-19 alone may not be too surprising. The CD8+ T cell tetramer studies were performed after *in vitro* expansions of CD8+ T cells with peptide stimulation – but are there naturally occurring clonal expansions of CD8+ T cells in active MIS-C?

We agree that identifying autoreactive CD8+ T cells targeting the same antigen as the autoantibodies is important. We also agree that naturally occurring clonal expansion provides stronger evidence of relevant cross-reactivity than expansions only observed after *in vitro* stimulation. To address this, in this revision, we provide new experiments in which we demonstrate natural clonal expansions of cross-reactive CD8+ T cells in MIS-C patients (see descriptions above, and the new Figure 5).

The following data would make this study of sufficient interest.

1. Evidence for natural clonal expansion of CD8+ T cells in the blood of MIS-C patients that cross-react with the relevant SNX8 nonamer.

We agree that identifying this would be highly important. As shown in the new Figure 5, we have now identified: 1) convergent TCR recognition from multiple clonal lineages, 2) clonal expansion *in vivo* and 3) true cross-reactive specificity *ex vivo*.

2. Showing that these clonally expanded CD8+ T cells can damage tissues that are known tissue targets in MIS-C

While identifying the tissue targets and sequelae of SNX8 autoreactivity are an important future direction, we felt that they are beyond the scope of the current study which has already made multiple significant findings as outlined above. Furthermore, the appropriate completion of these studies would require establishing an *in vivo* model or organoid system to study the relative effect of these CD8+ T cells on an entire organism under various priming conditions with and without prior SARS-CoV-2 (or other potentially yet-to-be discovered important predisposing factors in MIS-C) infection. Such a model does not currently exist, though we are working to develop such a system. Additionally, access to large banks of cryopreserved PBMCs would be needed in order to isolate sufficient numbers of *ex vivo* cross-reactive CD8+ T cells to study their effects *in vivo* because TCRs cloned into immortalized T cell lines would not be expected to recapitulate the function of the primary *in vivo* CD8 T cells. Despite having complete access to one of the largest MIS-C biorepositories of its kind, we have exhausted our supply of MIS-C PBMCs from patients with elevated SNX8 autoantibody levels performing the experiments in this paper. Not only do we not have sufficient samples to attempt these experiments, we are

unsure of whether there exists a sufficient global supply of samples. Nonetheless, we are continuing to explore all options to study the direct pathogenicity of these CD8 T cells in future studies.

1. Sabatino, J. J., Jr *et al.* Anti-CD20 therapy depletes activated myelin-specific CD8⁺ T cells in multiple sclerosis. *Proc. Natl. Acad. Sci. U. S. A.* **116**, 25800–25807 (2019).
2. Culina, S. *et al.* Islet-reactive CD8⁺ T cell frequencies in the pancreas, but not in blood, distinguish type 1 diabetic patients from healthy donors. *Sci Immunol* **3**, (2018).
3. Elong Ngonu, A. *et al.* Frequency of circulating autoreactive T cells committed to myelin determinants in relapsing-remitting multiple sclerosis patients. *Clin. Immunol.* **144**, 117–126 (2012).
4. Harndahl, M. *et al.* Peptide-MHC class I stability is a better predictor than peptide affinity of CTL immunogenicity. *Eur. J. Immunol.* **42**, 1405–1416 (2012).

Reviewer Reports on the First Revision:

Referees' comments:

Referee #1 (Remarks to the Author):

The authors have addressed all of my comments.

Referee #2 (Remarks to the Author):

The new data significantly strengthen the manuscript. Particularly relevant is the direct ex vivo isolation of TCRs with validated cross-reactivity.

Referee #3 (Remarks to the Author):

The specific autoantibody profiles, the CD4+ T cell responses and the additional data on epitope specific CD8+ T cell responses against the MQMPQGNPL peptide in SNX, including clonal expansion of T cells, are all convincing.